# XGRAG: A Graph-Native Framework for Explaining KG-based Retrieval-Augmented Generation

## Abstract

Graph-based Retrieval-Augmented Generation (GraphRAG) extends traditional RAG by using knowledge graphs (KGs) to give large language models (LLMs) a structured, semantically coherent context, yielding more grounded answers. However, GraphRAG reasoning process remains a "black-box", limiting our ability to understand how specific pieces of structured knowledge influence the final output. Existing explainability (XAI) methods for RAG systems, designed for text-based retrieval, are limited to interpreting an LLM's response through the relational structures among knowledge components, creating a critical gap in transparency and trustworthiness. To address this, we introduce **XGRAG**, a novel framework that generates causally grounded explanations for GraphRAG systems by employing graph-based perturbation strategies, to quantify the contribution of individual graph components on the model's answer. We conduct extensive experiments comparing XGRAG against RAG-Ex, an XAI baseline for standard RAG, and evaluate its robustness across various question types, narrative structures and LLMs. Our results demonstrate a 14.81% improvement in explanation quality over the baseline RAG-Ex across NarrativeQA, FairyTaleQA, and TriviaQA, evaluated by F1-score measuring alignment between generated explanations and original answers. Furthermore, XGRAG's explanations exhibit a strong correlation with graph centrality measures, validating its ability to capture graph structure. XGRAG provides a scalable and generalizable approach towards trustworthy AI through transparent, graph-based explanations that enhance the interpretability of RAG systems.

## 1 Introduction

The rise of Retrieval-Augmented Generation (RAG) (Lewis et al., 2021) has significantly improved large language models (LLMs) by grounding them on external knowledge. Early RAG systems relied on retrieving unstructured text chunks, but a new frontier has emerged with **GraphRAG** (Edge et al., 2024; Guo et al., 2024; Li et al., 2025), which leverages the rich relational structure of knowledge graphs (KGs). By representing information as entities and their relationships, GraphRAG systems can retrieve semantically coherent and contextually rich information, moving beyond simple keyword matching to understand the relationships between concepts.

However, this advance in retrieval has exposed a critical gap in **Explainable AI (XAI)** (Wu et al., 2025). While the graph structure makes retrieval more transparent, the LLM's subsequent reasoning process remains a "black-box." The central question *"How the model synthesizes information from various graph components to arrive at its final answer?"* is left unanswered by current tools. XAI methods for RAG, such as RAG-Ex (Sudhi et al., 2024) and RAGE (Rorseth et al., 2024), were developed for standard text-based RAG and are unsuitable for structured graph inputs. They fail to pinpoint the specific relationships and/or entities within the graph that were most influential, leaving users without a clear understanding of the model's decision-making process.

To address these limitations, we introduce **XGRAG**, a novel explainable AI framework tailored to Knowledge Graph-based RAG (GraphRAG) systems. XGRAG provides causally grounded explanations by applying graph-native perturbations to identify the most influential graph components

(nodes and edges) that contribute to an LLM's answer. The practical applications of XGRAG are numerous. In high-stakes domains such as medicine or finance, where the cost of an incorrect or unfaithful answer is high, our framework can be used to audit the model's reasoning process, ensuring that its conclusions are based on valid evidence. For developers, XGRAG serves as a powerful debugging tool, allowing them to pinpoint the specific knowledge graph components that may be leading to incorrect answers or hallucinations. Finally, for end-users, it fosters trust and transparency by providing a clear and understandable justification for the model's output, turning **the black box** of the LLM's reasoning into a transparent and auditable process. Unlike prior perturbation-based XAI methods for standard RAG, where perturbation operates solely at the text level, our work directly manipulates knowledge graphs. XGRAG offers a novel perspective on attributing the reasoning process of LLMs, by leveraging semantically coherent relationships between information units. This distinction between our graph-native approach and traditional text-based methods is illustrated in Figure 1.

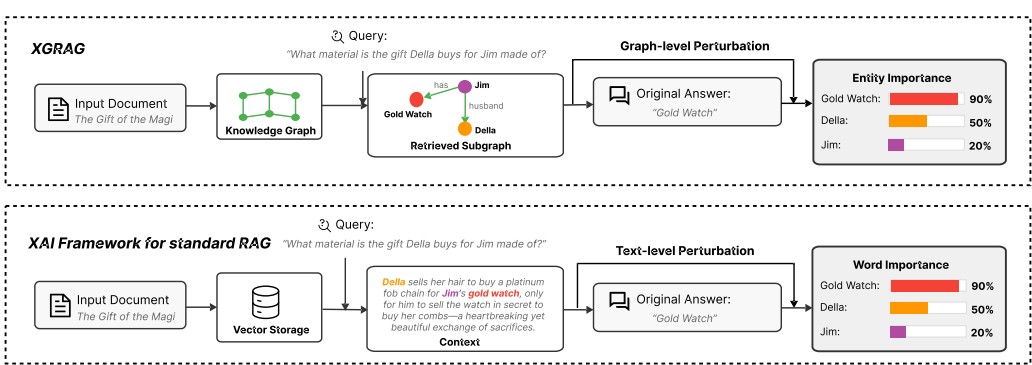

Figure 1: XGRAG vs. XAI framework for text-based RAG. Standard approaches (bottom) perturb unstructured text retrieved from a vector store to assess importance. Our framework XGRAG (top) operates on a KG, perturbing subgraphs to identify the key graph components for the LLM's answer.

Our key contributions are: (1) a novel XAI framework using graph-based perturbation to quantify the influence of graph components on LLM responses; (2) experiments showing that XGRAG outperforms text-based baseline RAG-EX, generalizes across question types, narrative complexities, and open-source LLMs, and aligns with graph centrality measures; and (3) ablation studies confirming the value of entity deduplication and clarifying the contributions of each graph perturbation strategy.

## 2  BACKGROUND AND RELATED WORK

**Retrieval-Augmented Generation.** RAG (Lewis et al., 2021) is a framework that enhances LLMs by integrating external knowledge retrieval into the generation process. It retrieves relevant documents from a knowledge base and conditions the generation on retrieved context using an LLM (Han et al., 2024). This approach improves factual accuracy and reduces hallucinations without retraining the model (Gao et al., 2024).

Still, RAG systems often fail to answer complex questions that require synthesizing information from diverse sources, such as identifying overarching themes across a dataset (Edge et al., 2024). Therefore, recent advancements in RAG have explored integrating structured data like KGs to improve the relevance and interpretability of retrieved context (Xu et al., 2024; Wang et al., 2025). GraphRAG (Edge et al., 2024) is a prominent example that leverages the relational structure of graphs to guide retrieval based on entity relationships and graph topology, rather than relying solely on lexical similarity, which often misses semantic context. This enables the system to retrieve context that is not only topically relevant but also semantically coherent, as related entities are connected through meaningful paths in the graph (Edge et al., 2024; Han et al., 2025). However, GraphRAG's reliance on full graph reconstruction and traversing can introduce scalability and efficiency challenges, especially when dealing with large or frequently updated datasets (Guo et al., 2024).

Building on GraphRAG, LightRAG (Guo et al., 2024) introduces a more efficient and flexible retrieval. Its key innovation is a dual-level retrieval: the low-level component targets specific entities and relationships for fine-grained queries, while the high-level component enables broader knowledge discovery. This design improves contextual relevance and coverage. Compared to GraphRAG, LightRAG is more efficient by integrating graph structures with vector-based similarity search and supporting incremental updates without full graph reconstruction. These features make LightRAG effective for tasks needing both entity-level precision and broader semantic understanding.

**Structured Explainability in RAG Systems: From Token-Level to Graph-Level Reasoning.**
Explainability in RAG systems remains a central challenge (Rorseth et al., 2024; Wu et al., 2025). Traditional methods such as Chain-of-Thought reasoning (Wei et al., 2023; Bilal et al., 2025) expose intermediate steps, but are often heuristic and lack causal grounding. To address this, RAG-Ex (Sudhi et al., 2024) introduced a model-agnostic, perturbation-based framework that identifies critical tokens in the retrieved context. By removing words or sentences and observing changes in the output, RAG-Ex uncovers causal relationships between context and answer. However, this approach applies perturbations broadly across the entire context, without focusing on the most meaningful elements, leading to higher computational cost and lower efficiency (Balanos et al., 2025).

Building on this idea, KGRAG-Ex (Balanos et al., 2025) applies perturbation at the graph level, removing nodes, edges, or paths to generate meaningful, causally grounded explanations than token-based methods. However, it lacks a dedicated evaluation pipeline for explanation quality. This limitation undermines the plausibility of the explainability results, as there is no objective basis for evaluating or trusting the explanations produced. XGRAG addresses this by combining graph-native perturbations with systematic evaluation, identifying key graph components and quantifying explanation quality with metrics reflecting the model's reasoning. Furthermore, unlike KGRAG-Ex, XGRAG fully leverages the superior retrieval and graph construction strategies of state-of-art GraphRAG frameworks, further enhancing both the scalability and effectiveness of our explainability pipeline.

## 3 METHODOLOGY

### 3.1 SYSTEM ARCHITECTURE

Our framework augments a standard GraphRAG pipeline (Edge et al., 2024) with a perturbation-based explanation layer to generate query-specific importance for graph components. The system architecture, depicted in Figure 2, is composed of four core modules: a GraphRAG backbone, an Entity Deduplication module, a Perturber, and an Explainer.

The process begins with the GraphRAG backbone constructing a global knowledge graph $G$ from the source documents. When a user submits a query $q$, the backbone retrieves a relevant subgraph, which we denote as $G_{ret}$. This subgraph is then passed to the Entity Deduplication module, which cleans the graph by merging semantically equivalent entities, resulting in a consolidated subgraph, $G_{dedup}$. This cleaned subgraph serves as the primary context for explanation. The generator of the backbone $g$, generates a baseline answer, $a_0 = g(G_{dedup})$. The Perturber then modifies $G_{dedup}$ by manipulating its nodes or edges, creating a set of counterfactual subgraphs, $G'_{dedup}$. For each counterfactual subgraph, the backbone is invoked again to produce a new, counterfactual answer, $a_p = g(G'_{dedup})$. Finally, the Explainer measures the semantic shift between each $a_p$ and the baseline answer $a_0$. A significant change in the answer indicates that the perturbed graph unit was highly influential in the model's reasoning for that specific query.

**GraphRAG Backbone.** Our framework is built upon a GraphRAG (Edge et al., 2024) backbone, which performs three key tasks: (1) constructing a global knowledge graph, $G = (V, E)$, from source documents during indexing; (2) retrieving a query-relevant subgraph, $G_{ret}$, for a given query during retrieval; and (3) synthesizing an answer from the retrieved context during generation.

The GraphRAG implementation from (Edge et al., 2024) is computationally expensive and thus unsuitable for our use case, as our perturbation-based method requires numerous pipeline executions for a single query, making computational efficiency a critical concern. To address this efficiency concern, we adopt LightRAG (Guo et al., 2024) as our backbone. Its lightweight design reduces the computational overhead, making our multi-run perturbation analysis feasible.

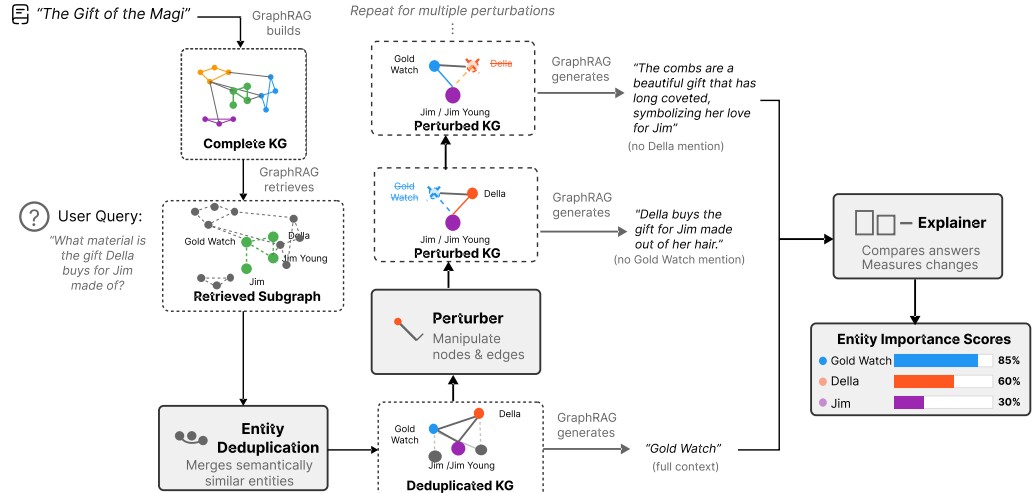

Figure 2: The XGRAG architecture. The GraphRAG backbone retrieves a subgraph, which is then deduplicated and perturbed. The Explainer module measures the semantic shift in the backbone's generated answers to score the causal importance of each graph component.

Beyond speed, our downstream perturbation requires a clean entity knowledge base. While LightRAG employs a deduplication process before finalizing $G$ during the indexing, it relies on exact key matching, i.e., only entities with identical names will be consolidated. This approach will fail to merge semantically equivalent entities with different names, such as aliases or abbreviations (e.g., "Dr. Watson" and "Watson"). This limitation results in a fragmented KG where information about a single conceptual entity is scattered across multiple nodes. To address this, we introduce an entity deduplication module to merge these near-duplicates before the perturbation stage.

**Entity Deduplication.** The initial subgraph retrieved by the backbone, $G_{ret}$, often contains redundant or semantically equivalent entities. The presence of redundant entities can fragment information across multiple nodes, leading to a noisy evaluation of entity importance. To resolve these inconsistencies, our deduplication module identifies and merges such entities based on the semantic similarity of entity names. We formalize this process of transforming a retrieved subgraph $G_{ret} = (V_{ret}, E_{ret})$ into a deduplicated subgraph $G_{dedup} = (V_{dedup}, E_{dedup})$. In this formalization, an edge is represented as a triple $(u, l, v)$, where $u$ and $v$ are entities and $l$ is the textual label of the relation.

1. **Similarity Graph Construction.** We build an undirected similarity graph $G_{sim} = (V_{ret}, E_{sim})$, where an edge $(v_i, v_j) \in E_{sim}$ exists if the entities $v_i, v_j \in V_{ret}$ share the same type and the cosine similarity of their name embeddings, generated by an embedding model $\mathcal{E}$, exceeds a predefined threshold $\theta_{sim}$:

$$(v_i, v_j) \in E_{sim} \iff \text{type}(v_i) = \text{type}(v_j) \land \text{sim}(\mathcal{E}(\text{name}(v_i)), \mathcal{E}(\text{name}(v_j))) \geq \theta_{sim}$$

2. **Clustering.** We find the connected components of $G_{sim}$, which partitions the set of subgraph entities $V_{ret}$ into a set of disjoint clusters $\mathcal{C} = \{C_1, C_2, \ldots, C_k\}$. Each cluster $C_i$ represents a group of semantically equivalent entities.

3. **Canonical Representative Selection.** For each cluster $C_i$, we select a canonical representative $v_i^*$ as the entity with the highest degree within the context subgraph $G_{ret}$.

$$v_i^* = \arg\max_{v \in C_i} \deg_{G_{ret}}(v)$$

4. **Graph Consolidation.** The final subgraph $G_{dedup} = (V_{dedup}, E_{dedup})$ is built. The new vertex set is the set of canonical representatives, $V_{dedup} = \{v_i^* \mid C_i \in \mathcal{C}\}$. A mapping $\phi : V_{ret} \to V_{dedup}$ sends each entity $v \in C_i$ to its representative $v_i^*$. The new edge set $E_{dedup}$ is formed by remapping original edges from $E_{ret}$ and removing resultant self-loops:

$$E_{dedup} = \{(\phi(u), l, \phi(v)) \mid (u, l, v) \in E_{ret} \land \phi(u) \neq \phi(v)\}$$

The textual descriptions of cluster entities are merged and assigned to the canonical representative.

**Perturber.** Our perturber adapts the *"Perturb-Generate-Compare"* scheme from RAG-Ex (Sudhi et al., 2024) to graph contexts. We apply this methodology by systematically manipulating the deduplicated subgraph $G_{dedup}$. For a given subgraph $G_{dedup}$ and a component $p \in V_{dedup} \cup E_{dedup}$, a perturbation creates a counterfactual graph $G'_{dedup}$. For perturbations like node or edge removal, this operation is a set difference: $G'_{dedup} = G_{dedup} \setminus \{p\}$. A complete list of our perturbation strategies is provided in Table 1.

Table 1: Graph-based perturbation strategies and their formalizations.

| Strategy | Description | Formalization |
|---|---|---|
| **Node Removal** | Removes a node $v$ and its incident edges to test the model's reliance on the entity. | $V'_{dedup} = V_{dedup} \setminus \{v\}$ 
 $E'_{dedup} = E_{dedup} \setminus$ 
 $\quad\quad \{(u, l, w) \in E_{dedup} \mid u = v \vee w = v\}$ |
| **Edge Removal** | Deletes an edge $e = (u, l, v)$ while keeping its endpoints to isolate the relationship's importance. | $V'_{dedup} = V_{dedup}$ 
 $E'_{dedup} = E_{dedup} \setminus \{e\}$ |
| **Synonym Injection** | Replaces an entity's name with a synonym to assess sensitivity to lexical variations. | For an entity $v$, a new entity $v'$ is created where $\text{name}(v') = \text{syn}(\text{name}(v))$. The graph is updated as: $V'_{dedup} = (V_{dedup} \setminus \{v\}) \cup \{v'\}$ 
 $E'_{dedup} = \{(\phi_v(u), l, \phi_v(w)) \mid (u, l, w) \in E_{dedup}\}$ 
 where $\phi_v(x)$ maps $v$ to $v'$ and other nodes to themselves. |

After perturbation, each of these counterfactual subgraphs is passed to the backbone's generator to produce a perturbed answer, which allows the Explainer to measure the causal impact.

**Explainer.** The Explainer module quantifies the causal influence of each graph component by measuring the semantic shift it causes in the model's output. Following RAG-Ex (Sudhi et al., 2024), we define the importance of a component $p$ as the semantic distance between the baseline answer $a_0$ from the original subgraph ($G_{dedup}$) and the counterfactual answer $a_p$ from the perturbed subgraph ($G'_{dedup}$). A larger distance signifies greater importance. We formalize this calculation using the generator function $g$ and the complement of cosine similarity:

$$\text{Imp}(p) = 1 - \text{sim}(a_0, a_p), \quad \text{where } a_0 = g(G_{dedup}), a_p = g(G'_{dedup})$$

To assess the contribution of each perturbation, we normalize importance scores across all perturbed units $P$ for a query. The normalized importance score $\text{Imp}_{\text{norm}}(p)$ for a unit $p \in P$ is calculated as:

$$\text{Imp}_{\text{norm}}(p) = \frac{\text{Imp}(p)}{\max_{p' \in P} \text{Imp}(p')}$$

This process ensures that the most influential unit for any given query receives a score of 1, while all other units are scored proportionally. A score near 1 signifies high influence, whereas a score near 0 suggests a minimal impact. This normalized score provides a clear and consistent measure of each graph component's relative importance in the model's reasoning process.

## 4 EXPERIMENTS AND RESULTS

### 4.1 DATASET AND GRAPH CONSTRUCTION

We evaluate our framework on three datasets which are focused on question-answering over long-form documents: NarrativeQA (Kočiský et al., 2017), FairyTaleQA (Xu et al., 2022) and TriviaQA (Joshi et al., 2017). NarrativeQA consists of stories from books and movie scripts, with questions designed to assess deep narrative understanding. The FairyTaleQA dataset provides a collection of fairy tales with question-answer pairs aiming to evaluate comprehension of narrative structures and moral reasoning. Finally, TriviaQA is a large-scale QA dataset featuring questions from trivia domains, it is designed to test factual and contextual understanding. For our experiments, we curated an evaluation set by selecting representative stories that cover different categories. The questions associated with these stories form our test set.

**Story Classification.** To evaluate our framework's performance across different narrative structures, we first categorize the stories individually, based on their content and complexity. This allows us to test the robustness of our explanation method on texts ranging from straightforward plots to narratives rich in dialogue or abstract themes. We define three categories: **Simple Narrative** stories feature linear plots; **Complex Plot** stories involve multiple subplots or a large cast; and **Abstract Concepts** stories explore philosophical themes.

**Question Classification.** Complementing the story-level analysis, we also classify questions to analyze performance based on cognitive complexity. Each query is categorized based on its leading interrogative pronoun, which serves as a proxy for its cognitive level according to a simplified version of Bloom's Taxonomy (Bloom et al., 1956). We distinguish between two types: **Factual Recall** questions are lower-order queries requiring the retrieval of explicit facts (e.g., starting with *What, Who, Where*), while **Inferential Reasoning** questions are higher-order queries that demand the synthesis of information to understand causality (e.g., starting with *Why, How*).

**Graph Construction.** The KG for each document collection was constructed using the GraphRAG backbone, whose LLM-based entity and relationship extraction pipeline processed the raw text to build a structured KG serving as the knowledge source for the system.

## 4.2 EXPERIMENTAL SETUP

**Models.** Our framework is designed to be model-agnostic. To demonstrate this, we evaluated its performance with a diverse set of open-source LLMs and embedding models. These include `gemma3-4b` (Kamath et al., 2025), `mistral-7b` (Jiang et al., 2023), `deepseek-r1-7b` (Marjanović et al., 2025), `llava-7b` (Liu et al., 2023), and `llama3.1-8b` (Grattafiori et al., 2024). All open-source models were run locally via Ollama, with which LightRAG seamlessly integrates. For consistency across experiments, each LLM handles both answer generation and graph construction, and `nomic-embed-text` (Nussbaum et al., 2024) was used for all embedding generation.

**Baseline.** To demonstrate the value of our graph-specific explanation approach, we compare it against the baseline **RAG-Ex** (Sudhi et al., 2024), which applies text-level perturbations to explain vector-based RAG systems. This allows us to isolate the benefits of our graph-native approach.

**Ground Truth.** To establish a reproducible ground truth for our evaluation, we adopt a computational approach that approximates human intuition about relevance. The core assumption is that graph components semantically similar to the final answer are the most relevant pieces of evidence. For a given query $q$, we first obtain the model's baseline answer, $a_0$. Then, for each graph unit $p$ (a node or an edge) in the retrieved context, we compute a relevance score, $\text{rel}(p) = \text{sim}(a_0, p)$. This relevance score serves a dual purpose in our ground truth definition. First, it provides a **ranked list** of all graph components, ordered from most to least relevant. This ranking is used as the ground truth for ranking-based metrics. Second, by applying a relevance threshold $\theta_r$, we create a **binary classification** for each component, which is used for classification-based metrics. A graph unit $p$ is labeled as a positive ground truth sample if its relevance score exceeds this threshold (i.e., $\text{rel}(p) > \theta_r$). This method creates a "golden set" of attributions for each query $q$. For instance, if the query is *"What do Lucie and Mrs. Tiggy-Winkle set off to do?"* and the answer is *"They set off to wash clothes."*, a node representing *"Clothes"* and an edge like *"(Mrs. Tiggy-Winkle, has occupation, washerwoman)"* would score highly and be included as positive ground truth samples.

## 4.3 EVALUATION

We evaluate our framework using a comprehensive set of metrics that measure performance from three key perspectives: explanation accuracy, ranking quality, and graph structural alignment.

**Explanation Accuracy.** We treat explanation generation as a binary classification task where each graph unit is classified as either "important" or "not important." To obtain these binary predictions, we use the normalized importance scores, $\text{Imp}_{norm}(p)$, generated for each graph component $p$. A component is classified as "important" if its score exceeds an importance threshold $\theta_{imp}$ (i.e., $\text{Imp}_{norm}(p) > \theta_{imp}$). We then evaluate the accuracy of these predictions against the ground truth using the F1-score, which is the harmonic mean of precision and recall.

**Ranking Quality.** While F1-score provides a holistic view of classification accuracy, in practice, users are often most interested in the top few pieces of evidence that justify an answer. Since most graph units in a retrieved context are less important, we place a strong emphasis on evaluating whether our framework can reliably identify and rank the most critical nodes and edges. The normalized importance scores $\text{Imp}_{norm}(p)$ induce a ranking of all components $p \in V_{sub} \cup E_{sub}$. We evaluate this ranking using two standard metrics:

*Mean Reciprocal Rank (MRR).* We use Reciprocal Rank (RR) because, in practice, we only care about the rank of the single most important piece of evidence. For each query, we calculate this as $\frac{1}{\text{rank}_i}$, where $\text{rank}_i$ is the position of this single ground truth item in the predicted list. Mean Reciprocal Rank (MRR) (Craswell, 2009) is the average of these scores over all queries $Q$, where higher values indicate that the framework consistently ranks the most critical evidence at the top.

*Precision at k% (P@k%).* While MRR is effective for the top-ranked item, its score can be inflated by smaller context sizes, making comparisons less reliable. To address this, we use P@k%, a scale-invariant metric that measures precision within the top k% of predictions:

$$\text{P@k\%} = \frac{|\{\text{Top-k\% Predicted}\} \cap \{\text{Top-k\% Ground Truth}\}|}{N_k}$$

where $N_k$ is the number of items corresponding to the top k% of the list. In our experiments, we evaluate P@k% for k values of 10, 30, and 50, to assess performance across different importance tiers, from the most critical items (top 10%) to a broader portion of the context (top 50%).

**Graph Structural Alignment.** To validate that our explainer's importance scores align with the structural properties of the graph, we measure the correlation between node importance and graph centrality. We hypothesize that structurally important nodes should receive higher importance scores. We evaluate this using two standard centrality measures, Degree Centrality and PageRank.

*Degree Centrality.* A measure where the centrality of a node $v \in V$ is its degree, $C_D(v) = \deg(v)$.

*PageRank.* A more sophisticated measure that assigns importance based on the quantity and quality of incoming links (Page et al., 1999). The PageRank score for a node $v$ is defined recursively:

$$PR(v) = \frac{1-d}{N} + d \sum_{u \in M(v)} \frac{PR(u)}{L(u)}$$

where $M(v)$ is the set of nodes linking to $v$, $L(u)$ is the number of outbound links from node $u$, $N$ is the total number of nodes, and $d$ is a damping factor (typically 0.85).

For both centrality measures, we compute the **Spearman rank correlation coefficient ($\rho$)** (Daniel, 1990) between the ranking of nodes induced by our explainer's importance scores and the ranking induced by their centrality. Alongside the coefficient, we compute the **p-value** (Best & Roberts, 1975) to determine the statistical significance of the correlation. A statistically significant positive correlation, indicated by a low p-value (e.g., $< 0.05$), provides evidence that our graph-native explanation method captures structurally relevant information.

## 4.4 RESULTS

**Comparison with Baseline.** To demonstrate the value of our graph-native explanation approach, we compare it against the baseline **RAG-Ex** (Sudhi et al., 2024). To approach a fair comparison, we align the perturbation granularities: the baseline perturbs text at the word- and sentence-level, while our graph-native approach perturbs the graph at the corresponding node- and edge-level. To ensure a robust comparison, we evaluated both frameworks across all stories and question types from the test set. As shown in Table 2, our graph-native perturbations significantly outperform the baseline's text-based counterparts across all metrics at both levels. This performance gap underscores the fundamental advantage of our graph-native approach, which achieves superior explanation accuracy by perturbing semantically coherent graph components rather than unstructured text.

**Robustness to Data and Task Variations.** To demonstrate the robustness of our framework, we conducted further a fine-grained analysis across different question types (task variations) and narrative structures (data variations). First, we test robustness against cognitive complexity by analyzing performance on Factual Recall and Inferential Reasoning questions. As visualized in Figure 3, our framework outperforms the baseline on both types of questions. The advantage is most observed for *Inferential Reasoning* questions, where our model's ranking performance is significantly better with

Table 2: Main evaluation results comparing XGRAG against baseline RAG-Ex. All perturbations use a removal strategy, with the baseline operating on text at word- and sentence-level and our method operating on graph at node- and edge-level. The reported metrics are averaged across all story and question types.

| Method | Granularity | F1 | MRR | P@10% | P@30% | P@50% |
|---|---|---|---|---|---|---|
| **RAG-Ex (Baseline)** | word-level | 0.54 | 0.23 | 0.08 | 0.11 | 0.19 |
| | sentence-level | 0.34 | 0.61 | 0.35 | 0.42 | 0.54 |
| **XGRAG (Ours)** | node-level | **0.62** | **0.72** | **0.66** | **0.44** | **0.57** |
| | edge-level | 0.52 | 0.65 | 0.22 | 0.42 | 0.48 |

an MRR more than double and a P@10% over five times higher than the baseline. XGRAG proves exceptionally effective at identifying the critical evidence needed for complex reasoning tasks.

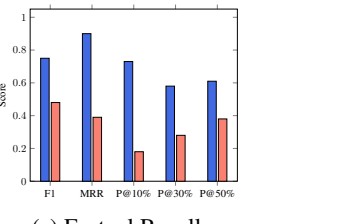 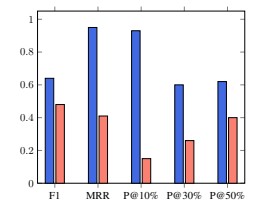 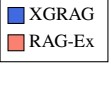

(a) Factual Recall        (b) Inferential Reasoning

Figure 3: Performance comparison on questions with different cognitive levels. These results compare XGRAG and RAG-Ex using node- and word-level removal strategy, respectively, applied to the *"Goldilocks and the Three Bears"* story with `llava-7b` model.

Second, we investigate performance consistency across different narrative structures. The results in Figure 4 demonstrates that XGRAG achieves superior performance compared to the baseline RAG-Ex across the three story types. Notably, the performance gap is most significant for "Simple Narrative" stories, which suggests that even in low-complexity scenarios, our graph-native method's ability to precisely target explicit relationships provides a substantial advantage over text-based approaches that struggle to isolate key facts from irrelevant context.

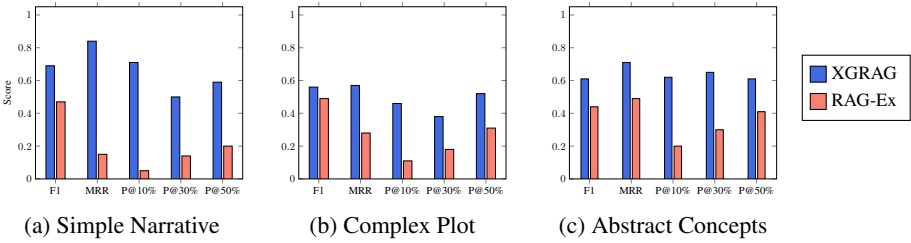

(a) Simple Narrative     (b) Complex Plot     (c) Abstract Concepts

Figure 4: Performance comparison of XGRAG against baseline RAG-Ex across different narrative structures. These results are generated using word- and node-level removal strategy with the `llava-7b` model, demonstrate our method's consistent advantage across all story types.

**Generalization Across LLMs.** To validate the open-source, model-agnostic **generalization** capabilities of our framework, we evaluated its performance across several open-source LLMs of varying sizes. As shown in Table 3, XGRAG shows broad compatibility for all models. Despite minor variations, both F1 and MRR scores remain robust, indicating that our graph-native perturbation and evaluation logic is not overfitted to any specific open-source LLM.

**Graph Structural Alignment.** Having established the framework's robust performance and generalization capabilities, we now focus on an intrinsic evaluation of its graph-native properties. A key hypothesis unique to graph-native approaches is that structurally important nodes should receive higher importance scores, since perturbing these nodes would result in significant contextual loss. To access whether our explainer's importance scores implicitly reflect the underlying graph structure, we analyzed their correlation with standard node centrality metrics.

Specifically, we compute the Spearman rank correlation coefficient between the importance scores and centrality measures, filtering for statistically significant results ($p < 0.05$). As shown in Figure 5, strong correlations were most observed for both metrics among statistically significant results,

Table 3: Performance of XGRAG across different open-source LLMs. The framework maintains high F1 and MRR scores, demonstrating its model-agnostic nature and strong generalization. These results were generated using the node-level removal on the *"Goldilocks and the Three Bears"* story.

| LLM | F1-Score | MRR | P@10% | P@30% | P@50% |
|---|---|---|---|---|---|
| gemma3-4b | 0.44 | 0.53 | 0.20 | 0.52 | 0.54 |
| llava-7b | **0.71** | **1.00** | **1.00** | **0.67** | 0.67 |
| mistral-7b | 0.62 | **1.00** | **1.00** | 0.59 | **0.75** |
| deepseek-r1-7b | 0.58 | 0.50 | 0.25 | 0.38 | 0.56 |
| llama3.1-8b | 0.46 | 0.79 | 0.50 | 0.50 | 0.61 |

particularly with Degree Centrality. This demonstrates our explainer's ability to capture graph's topology, effectively identifying nodes that are not only semantically relevant but also structurally central, both of which are crucial for LLM's answer generation.

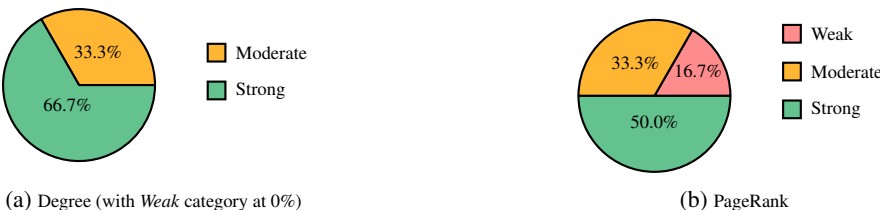

(a) Degree (with *Weak* category at 0%)    (b) PageRank

Figure 5: Breakdown of correlation strengths for statistically significant results ($p < 0.05$). Correlation strength is categorized as Weak ($|\rho| \leq 0.4$), Moderate ($0.4 < |\rho| < 0.6$), and Strong ($|\rho| \geq 0.6$).

## 4.5 SCALABILITY AND EFFICIENCY ANALYSIS

To quantify the computational cost of XGRAG, we refer to the comparative analysis performed in the LightRAG paper (Guo et al., 2024). Although their evaluation was conducted on a legal dataset, the results are illustrative of the performance difference between GraphRAG and LightRAG. As reported by Guo et al. (2024), the GraphRAG implementation incurs substantial overhead during the retrieval phase compared to LightRAG. The cost for a single query is summarized in Table 4.

Table 4: Comparative cost of a single query retrieval, based on the analysis by Guo et al. (2024). $C_{max}$ is the maximum tokens per API call.

| Backbone | Token Load | API Calls |
|---|---|---|
| GraphRAG | $\approx 610,000$ | $\approx 610,000/C_{max}$ |
| LightRAG | $< 100$ | 1 |

The total cost of generating an explanation in XGRAG, represented as $C_{XGRAG}$, can be formulated as:

$$C_{XGRAG} = C_{baseline} + \sum_{p \in P} C_p \approx N_p \times C_{invoke}$$

where $N_p = N_{entities} \vee N_{edges}$, which is strategy-dependent, and $C_{invoke}$ is the cost of a single retrieval backbone invocation. By substituting $C_{invoke}$ with the costs reported by Guo et al. (2024), the advantage becomes clear. A hypothetical implementation of XGRAG on GraphRAG backbone would be computationally infeasible:

$$C_{XGRAG-GR} \propto N_p \times (\text{100s of API calls} + 10^5\text{-}10^6 \text{ tokens})$$

Conversely, by building on LightRAG, the cost remains manageable:

$$C_{XGRAG-LR} \propto N_p \times (\text{1 API call} + 10^2 \text{ tokens})$$

This analysis demonstrates that while our perturbation strategies multiplies the cost of a single query, the efficiency of LightRAG is the key enabling factor that makes XGRAG a practical and scalable solution for explaining graph-based RAG. While scalability to extremely large Knowledge Graphs remains a topic for future work, our approach is demonstrably efficient for the common use cases evaluated in this paper.

Table 5: Ablation study on the *Entity Deduplication* module. Performance is compared using the node-level removal perturbation strategy on the story *"Goldilocks and the Three Bears"*.

| Configuration | F1 | MRR | P@10% | P@30% | P@50% |
|---|---|---|---|---|---|
| **XGRAG (Full Framework)** | **0.71** | **1.00** | **1.00** | **0.67** | **0.67** |
| - w/o Entity Deduplication | 0.41 | 0.66 | 0.50 | 0.36 | 0.52 |

## 4.6 ABLATION STUDIES

**Impact of Entity Deduplication.** The entity deduplication module merges synonymous entity nodes (e.g., *"Gold Watch"*, *"Watch"*, and *"The Watch"*) into a single canonical representation. We compared our framework against a version where this module is disabled, leaving synonymous entities as distinct nodes in the graph. As shown in Table 5, removing the deduplication module leads to a notable degradation in performance across all metrics. The results confirm that deduplication is a critical preprocessing step for building a **robust** knowledge graph. Without it, the graph becomes fragmented, scattering information about a single entity across multiple nodes and undermining the **robustness** of the explanation process.

**Comparison of Perturbation Strategies.** Beyond entity deduplication, our framework incorporates distinct perturbation strategies: node removal, edge removal, and synonym injection. This study evaluates the effectiveness of each perturbation strategy in identifying important graph components. Each strategy is applied independently, and its performance is assessed using the previously employed set of evaluation metrics. As shown in Table 6, **Node Removal** achieves best results across most metrics, confirming it as the most effective strategy for identifying the most critical graph components. This suggests that the presence or absence of an entire entity serves as a strong indicator of its causal importance. Removing a node, along with all its incident edges, results in the greatest information loss, thereby producing the strongest causal signal when a component is important.

Table 6: Performance comparison of the perturbation strategies from XGRAG. The results are based on experiments using `llava-7b` with *"Goldilocks and the Three Bears"* story as input.

| Perturbation Strategy | F1 | MRR | P@10% | P@30% | P@50% |
|---|---|---|---|---|---|
| Node Removal | **0.71** | **1.00** | **1.00** | **0.67** | 0.67 |
| Edge Removal | 0.32 | 0.84 | 0.40 | 0.30 | 0.25 |
| Synonym Injection | 0.23 | 0.56 | 0.20 | 0.54 | **0.70** |

## 5 CONCLUSION

By leveraging graph-native perturbation strategies, **XGRAG** generates fine-grained explanations that identify the most influential nodes and edges contributing to an LLM's response. Experiments across NarrativeQA, FairyTaleQA, and TriviaQA demonstrate that XGRAG outperforms text-based baseline RAG-Ex across all metrics: explanation accuracy, ranking quality, and alignment with graph structural properties. Moreover, XGRAG shows strong robustness and generalization, maintaining high performance across diverse question types, narrative complexities, and multiple open-source LLMs.

**Limitations and Future Work.** While XGRAG advances explainability for GraphRAG systems, several limitations remain. Our evaluation relies on semantic similarity as the primary metric for faithfulness assessment, although scalable, may introduce inherent biases. Future work should prioritize more robust ground-truth methodologies, such as semi-automated evaluation frameworks leveraging advanced LLMs as judges and human-annotated datasets with fine-grained relevance scores. Additionally, our current evaluation is constrained to local LLMs and English narrative datasets. Extending this work to incorporate larger proprietary LLMs, multilingual and domain-specific knowledge graphs would address emerging challenges in both explanation quality and system scalability.

## ETHICS STATEMENT

This research adheres to the ICLR Code of Ethics. Our work is foundational research in Explainable AI (XAI) with the primary goal of increasing the transparency and trustworthiness of GraphRAG systems. This work uses publicly available datasets, including NarrativeQA, FairyTaleQA, and

TriviaQA, which are derived from published materials (e.g., books, fairy tales, movie scripts, and curated trivia). We acknowledge that while our framework is designed to be neutral, the underlying LLMs and source data may contain inherent biases. A positive ethical implication of our work is that XGRAG can be used as a tool to audit and identify such biases by making the model's reasoning process more transparent. We believe that by providing causal explanations, our framework contributes positively to the responsible development and deployment of advanced AI systems.

## REPRODUCIBILITY STATEMENT

We have made every effort to ensure the reproducibility of our work. The core architecture of our XGRAG framework, including the entity deduplication and perturbation strategies, is detailed in the Methodology section. The complete experimental setup, including the specific stories and questions used from the public NarrativeQA, FairyTaleQA, and TriviaQA datasets, model versions, and all hyperparameter settings, is described in the Experiments and Results section. The source code for our framework, along with the scripts required to reproduce all experiments and generate the figures presented in this paper, will be made publicly available upon publication.

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

## APPENDIX

## LLM USAGE DISCLOSURE

In accordance with ICLR policy, we disclose the use of Large Language Models (LLMs) at multiple stages of this research.

**LLMs as a Core Research Component.** LLMs are integral to our research methodology. As detailed in the Methodology section, an LLM serves two primary functions within the XGRAG framework: (1) as the engine for knowledge graph construction from raw text, and (2) as the answer generator within the GraphRAG backbone, producing both baseline and counterfactual answers for our perturbation analysis. The specific models used are listed in the Experimental Setup section.

**LLMs as an Assisting Tool.** Beyond their role in the methodology, LLMs were used as assisting tools in the preparation of this work. For manuscript writing, an LLM was used to improve grammar, clarity, and style. For software implementation, an LLM served as a coding assistant for tasks such as debugging, refactoring, and polishing the Python code for the experimental pipeline. The core scientific claims, experimental design, and analysis of results were conceived and written **by the human authors**.

**Author Responsibility.** Following ICLR policy, the authors have reviewed and take full responsibility for all content in this submission, including the accuracy of claims and the correctness of any text or code potentially influenced by an LLM. It is important to note that LLMs were **not** used for the evaluation of the explanations themselves; our ground truth creation and evaluation metrics are based on a computational, **non-LLM-based** approach to ensure objective and reproducible results.

## A    VISUAL ANALYSIS

To provide an intuitive understanding of our framework's output, we present a visual analysis based on a representative example. Consider the query from our dataset: *"What material is the gift Della buys for Jim made out of?"*, and answer: *"Gold watch"*.

Our GraphRAG backbone retrieves a subgraph containing 21 nodes and 15 edges. As visualized in Figure 6, our XGRAG framework generates a precise explanation by assigning high importance scores to the core components of the answer. The node *"Della"* receives the highest score, and the node *"Gold Watch"* the second-highest score (the most relevant node for the query). The impact of these nodes is confirmed by their perturbations: removing *"Della"* causes the model to hallucinate, while removing *"Gold Watch"* leads to incorrect answer. This demonstrates our method's capacity to accurately identify the key entities that are causally responsible for generating the answer.

## B    TABLE OF NOTATIONS

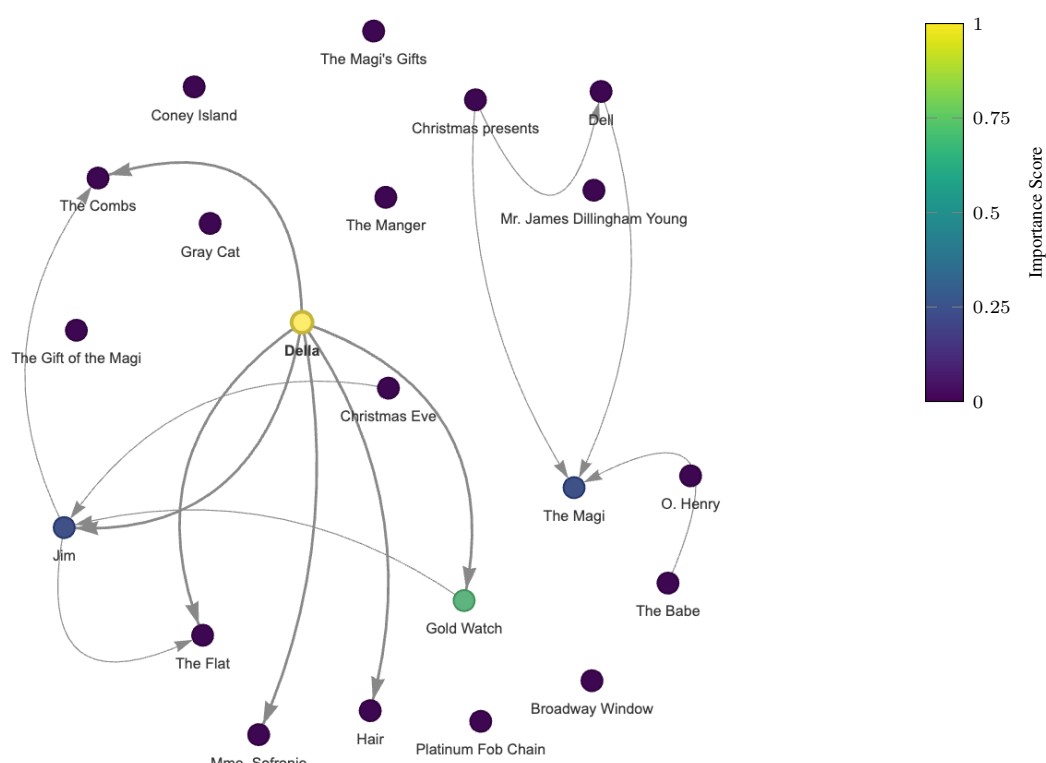

*Question: "What material is the gift Della buys for Jim made out of?"*

*Answer: "Gold Watch."*

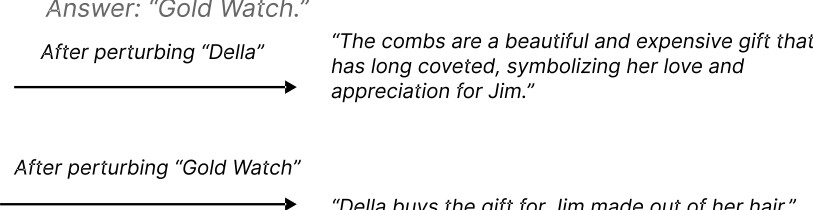

*After perturbing "Della"* → *"The combs are a beautiful and expensive gift that has long coveted, symbolizing her love and appreciation for Jim."*

*After perturbing "Gold Watch"* → *"Della buys the gift for Jim made out of her hair."*

Figure 6: Qualitative analysis for the query about Della's gift. The top image visualizes the retrieved subgraph, with node importance indicated by the colorbar. The bottom image shows the question and the model's generated answer.

Table 7: Summary of mathematical notations used in this paper.

| Symbol | Description |
|---|---|
| $q$ | A user query. |
| $p$ | A generic graph component (node or edge). |
| $l$ | The textual label or description of a relation (edge). |
| $f$ | The generator function of the GraphRAG backbone. |
| $\mathcal{E}$ | An embedding model. |
| $\mathrm{sim}(a, b)$ | Cosine similarity between the embeddings of items $a$ and $b$. |
| $G$ | The global knowledge graph. |
| $V, E$ | The set of nodes (entities) and edges (relations) in a graph. |
| $G_{ret}$ | The initial subgraph retrieved from $G$ for a query $q$. |

| Symbol | Description |
|---|---|
| $G_{dedup}$ | The deduplicated subgraph used as context for explanation. |
| $G'_{dedup}$ | A counterfactual subgraph created by perturbing $G_{dedup}$. |
| $G_{sim}$ | An undirected similarity graph used for entity deduplication. |
| $a_0$ | The baseline answer generated from the original subgraph $G_{dedup}$. |
| $a_p$ | The counterfactual answer generated from a perturbed subgraph $G'_{dedup}$. |
| $\text{Imp}(p)$ | The raw importance score of a graph component $p$. |
| $\text{Imp}_{\text{norm}}(p)$ | The normalized importance score of a component $p$. |
| $\theta_{imp}$ | The importance threshold for classifying a component as important. |
| $\text{Predicted}(p)$ | The binary predicted importance label for component $p$. |
| $a_q$ | The correct, ground truth answer for a query $q$. |
| $\text{rel}(p)$ | The relevance score of a component $p$ relative to the answer $a_q$. |
| $\theta_r$ | The relevance threshold for creating the binary ground truth set. |
| $\text{GroundTruth}(p)$ | The binary ground truth relevance label for component $p$. |
| $\theta_{sim}$ | The similarity threshold for merging two entities. |
| $\mathcal{C}$ | The set of disjoint entity clusters found during deduplication. |
| $v_i^*$ | The canonical representative entity for a cluster $C_i$. |
| $\phi$ | A mapping from an entity to its canonical representative. |
| $\text{rank}_i$ | The rank of the top ground truth item for query $i$. |
| $Q$ | The set of all queries in the evaluation set. |
| MRR | Mean Reciprocal Rank. |
| P@k% | Precision at k-percent. |
| $C_D(v)$ | Degree Centrality of a node $v$. |
| $PR(v)$ | PageRank score of a node $v$. |
| $d$ | The damping factor used in the PageRank calculation. |
| $\rho$ | The Spearman rank correlation coefficient. |

## C  DATASET DETAILS

This appendix provides further details on the datasets used for our evaluation. We curated evaluation sets from the **NarrativeQA** (Kočiský et al., 2017), **FairyTaleQA** (Xu et al., 2022), and **TriviaQA** (Joshi et al., 2017) datasets. The table below lists the documents and sample questions used to test our framework across various narrative and cognitive complexities.

Table 8: Evaluation Corpus Documents and Sample Questions.

| Story Category | Document Title | Question Type | Sample Questions |
|---|---|---|---|
| Simple Narrative | *"Goldilocks and the Three Bears"* | Factual | *"What did Goldenhair eat?"* |
| | | | *"Who lives in the house in the woods?"* |
| | | | *"Where do the bears go while the porridge cools?"* |
| | | Inferential | *"How does Goldenhair escape?"* |
| | | | *"Why did the bears leave their house?"* |

| Story Category | Document Title | Question Type | Sample Questions |
|---|---|---|---|
| | *"The Tale of Mrs. Tiggy-Winkle"* | Factual | *"What do Lucie and Mrs. Tiggy-Winkle set off to do?"* |
| | | | *"Who is well acquainted to Mrs. Tiggy-Winkle?"* |
| | | | *"What has Lucie lost?"* |
| | | Inferential | *"Why do Lucy and Tiggy-winkle set down the path?"* |
| | | | *"Why has Mrs. Tiggy-Winkle taken Lucie's things?"* |
| | *"The Straw, the coal and the bean story"* | Factual | *"What happened when one of the beans fell out and lay near the straw?"* |
| | | | *"Who lived in a certain village?"* |
| | | | *"Where did the straw, coal, and bean try to cross?"* |
| | | Inferential | *"How did the bean help out the straw?"* |
| | | | *"Why did the poor old woman collect a mess of beans?"* |
| | *"Golden boy Promotions"* | Factual | *"What major fight did Golden Boy Promotions promote on May 5, 2007?"* |
| | | | *"Who founded Golden Boy Promotions?"* |
| | | | *"Which US boxing world champion founded 'Golden Boy Promotions' in 2001?"* |
| | | Inferential | *"How did Golden Boy Promotions make history in 2006?"* |
| | | | *"Why did Golden Boy's mixed martial arts promotion with Affliction fold?"* |
| Complex Plot | *"The Tale of Samuel Whiskers"* | Factual | *"What ingredients do the rats cover Tom Kitten with?"* |
| | | | *"Who is the carpenter that came to help get Tom Kitten out of the attic?"* |
| | | | *"Where do the rats escape to after they are caught?"* |
| | | Inferential | *"How did Tom Kitten escape from the cupboard?"* |

| Story Category | Document Title | Question Type | Sample Questions |
|---|---|---|---|
| | | | *"Why did Tabitha put her children in the cupboard?"* |
| | *"The Adventure of the Dying Detective"* | Factual | *"What did Watson believe was wrong with Holmes?"* |
| | | | *"Who did Mr. Smith kill before?"* |
| | | | *"Where does Homes instruct Watson to go?"* |
| | | Inferential | *"How did Holmes feel when Watson touched the items in his room?"* |
| | | | *"Why did Watson hide behind a screen?"* |
| | *"An Occurrence at Owl Creek Bridge"* | Factual | *"What does Peyton own?"* |
| | | | *"What is Peyton's age?"* |
| | | | *"Where is Peyton going to be hanged?"* |
| | | Inferential | *"How is Peyton going to be executed?"* |
| | | | *"How does Farquhar escape?"* |
| | *"The coming of finn story"* | Factual | *"What was also called All Hallows' Eve?"* 
 *"Who was captain of all the Fians?"* |
| | | | *"Where did the king sit at supper?"* |
| | | Inferential | *"How did Allen burn Tara?"* |
| | | | *"Why was the king willing to give anything to Finn as a reward?"* |
| | *"Duke of Richmond"* | Factual | *"What happened to the first creation of the Dukedom of Richmond?"* 
 *"Where is the family seat of the Dukes of Richmond?"* |
| | | | *"Which West Sussex family seat is the home of the Dukes of Richmond and Gordon?"* |
| | | Inferential | *"How did the current Dukedom of Richmond originate?"* |

Table 8 – continued from previous page

| Story Category | Document Title | Question Type | Sample Questions |
|---|---|---|---|
| | | | "Why did the titles of the second creation of the Dukedom of Richmond become extinct?" |
| | "2007 Monaco Grand Prix" | Factual | "What issue caused Kimi Räikkönen to start the race from 16th place?" "Who won the Monaco Grand Prix in 2000?" |
| | | | "Where did the pre-race testing take place to simulate the Monaco circuit?" |
| | | Inferential | "How did Fernando Alonso secure pole position during qualifying?" |
| | | | "Why did the FIA investigate McLaren after the race?" |
| Abstract Concepts | "The Peach Blossom Spring" | Factual | "When was the Peach Blossom Spring written?" |
| | | | "What was the forest made of?" |
| | | | "What was the source of the river?" |
| | | Inferential | "How did the villagers react to the fisherman?" |
| | | | "Why were the villagers there?" |
| | "The Gift of the Magi" | Factual | "How much cash does Della have to spend on gifts?" |
| | | | "What material is the gift Della buys for Jim made out of?" |
| | | | "What are the two possessions that James and Della take pride in?" |
| | | Inferential | "How did Jim get the cash to buy the combs?" |
| | | | "Why did Della sell her hair?" |
| | "Grandmother Story" | Factual | "What does Grandmother look like?" |
| | | | "Who sits next to Grandmother in her memory?" |
| | | | "Where was the rose-tree planted?" |

Table 8 – continued from previous page

| Story Category | Document Title | Question Type | Sample Questions |
|---|---|---|---|
| | | Inferential | *"How did the children feel when they looked at Grandmother's corpse?"* |
| | | | *"Why does Grandmother smile at the rose?"* |

## D    ACHIEVING PUNCTUAL AND PRECISE EXPLANATIONS

A core challenge in perturbation-based explanation is ensuring that the model's response to a perturbation is both punctual and precise. A noisy or verbose counterfactual answer makes it difficult to isolate the causal impact of a single change. To address this, we implemented four key strategies to improve model performance and the reliability of our explanations.

### D.1    STRICT AND MINIMALIST PROMPTING

To achieve more punctual and precise results for our perturbation analysis, we refined the default answer generation prompt from LightRAG. The original prompt is designed for general-purpose RAG and includes instructions for formatting and incorporating external knowledge. Our refined version, in contrast, is stricter and more minimal, forcing the model to be concise and ground its answer strictly in the provided context. This is critical for isolating the causal impact of a single perturbation.

The table below shows a side-by-side comparison of the two prompts.

Table 9: Comparison of Answer Generation Prompts

| Original Response Rules | Refined Response Rules |
|---|---|
| *- Target format and length: {response_type}* | *- Target format and length: {response_type}* |
| *- Use markdown formatting with appropriate section headings* | *- Please respond in the same language as the user's question.* |
| *- Please respond in the same language as the user's question.* | *- Avoid varying the introductory sentence. Do not use alternatives like "According to..." or "From what we know..." — consistency is key.* |
| *- Ensure the response maintains continuity with the conversation history.* | *- If you don't know the answer, just say so.* |
| *- List up to 5 most important reference sources at the end under "References" section. Clearly indicating whether each source is from Knowledge Graph (KG) or Document Chunks (DC), and include the file path if available, in the following format: [KG/DC] file_path* | *- Do not make anything up. Do not include information not provided by the Knowledge Base.* |
| | *- Additional user prompt: {user_prompt}* |
| *- If you don't know the answer, just say so.* | |
| *- Do not make anything up. Do not include information not provided by the Knowledge Base.* | |
| *- Additional user prompt: {user_prompt}* | |

## D.2 REDUCED GENERATION TEMPERATURE

To further minimize randomness and creativity in the generated answers, we set the LLM's temperature hyperparameter to a low value of 0. This encourages the model to produce more deterministic and factual outputs based strictly on the provided context, which is essential for a stable perturbation analysis. A lower temperature reduces the likelihood of hallucination and ensures that the model's output is a direct function of the input context. As illustrated in Figure 7, a lower temperature directly improves the final F1-score of the explanations.

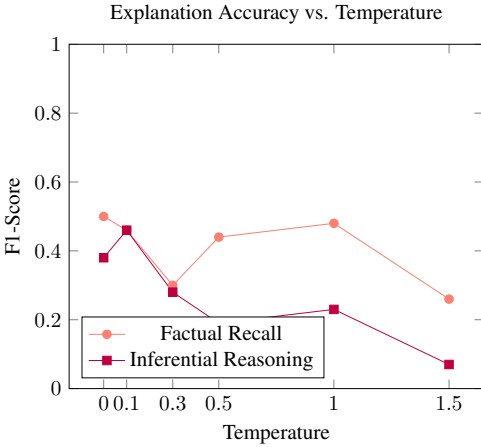

Figure 7: Impact of generation temperature on explanation accuracy. A low temperature (0.1) improves the final explanation accuracy (F1-Score).

## D.3 HYPERPARAMETERS

The experimental setup was configured with the following key parameters. For the 'Entity Deduplication' module, the similarity threshold $\theta_{sim}$ was set to 0.7. For 'Ground Truth' creation, the relevance threshold $\theta_r$ was set to 0.5. It is important to note that our evaluation is performed only on questions that can be answered, mitigating the risk of evaluating biased explanations. The importance threshold $\theta_{imp}$, used for binary classification in the 'Explanation Accuracy' evaluation, was also set to 0.5. For 'Ranking Quality', we evaluated Precision at k% for $k \in \{10, 30, 50\}$. Finally, the damping factor $d$ for the 'PageRank' calculation was set to the standard value of 0.85.

## D.4 ENTITY DEDUPLICATION SENSITIVITY ANALYSIS

This analysis determines the impact of varying $\theta_{sim}$ on entity deduplication performance and justify the threshold used in the experiment phase. Entity deduplication relies on a similarity threshold to decide whether two entities should be merged. The analysis in Figure 8 evaluates performance across a range of similarity thresholds from 0.0 to 1.0.

Our results show that performance remains relatively stable for low thresholds (0.0–0.3), with F1 scores around 0.828. However, mid-range thresholds (0.45–0.6) exhibit a noticeable drop in F1 (as low as 0.694). At higher thresholds (0.65), F1 increases to 0.794, and ranking metrics improve significantly. While $\theta_{sim} = 0.7$ does not achieve the highest F1 score overall, it consistently outperforms other thresholds in MRR and ranking overlap metrics (Top10%, Top30%, Top50%). This indicates superior ranking quality and retrieval consistency, which are essential for downstream tasks. Therefore, $\theta_{sim} = 0.7$ represents the best balance between precision, recall, and ranking performance, avoiding the mid-range dip and aligning with the region of maximum overlap.

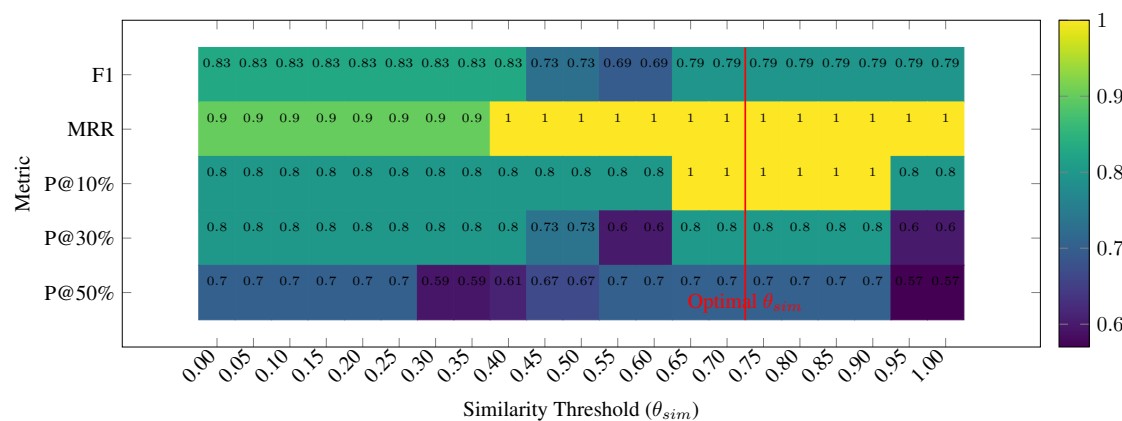

Figure 8: Sensitivity analysis of similarity threshold ($\theta_{sim}$) on entity deduplication performance. Metrics include F1, Mean Reciprocal Rank (MRR), and overlap at P@10%, P@30%, and P@50%. The results are based on experiments using `llava-7b` with *"Goldilocks and the Three Bears"* story as input.

## D.5 GRAPH-ONLY CONTEXT

Standard RAG systems often provide both structured data (like a graph) and unstructured text chunks as context. However, for explaining a GraphRAG system, the unstructured text can introduce noise, as it may contain redundant or conflicting information. Therefore, we configured our framework to provide only the deduplicated subgraph ($G_{dedup}$) as context to the answer generation model. This ensures that the explanation is based purely on the structured knowledge the model is reasoning over, eliminating noise from raw text chunks.

