# OpenReview forum: "XGRAG: A Graph-Native Framework for Explaining KG-based Retrieval-Augmented Generation"
_ICLR.cc/2026/Conference — Submitted to ICLR 2026_

### Official Review · Reviewer_tvRz · 2025-10-30

**Soundness:** 2
**Presentation:** 3
**Contribution:** 2
**Rating:** 2
**Confidence:** 4

**Summary:**

The paper introduces an explainability method for GraphRAG that assigns importance scores to entities and relations by perturbing them.

**Strengths:**

S1. The paper is well-written and the figures are clear, making the proposed method easy to follow and understand.

**Weaknesses:**

W1. The connection of this work to "explainability" is tenuous. The output consists of importance scores for entities and relations, but these scores do not truly explain the inner workings of the llms or why it generated a specific answer. For example, in the case study from Figure 1, knowing that the three entities (Gold Watch, Delta, Jim) all have non-zero importance scores does not explain how the model utilized these entities to formulate its response. In my view, this is more akin to "importance attribution" or "evidence identification" rather than genuine explainability.

W2. The experimental evaluation is insufficient. The entire study is conducted on a single dataset, NarrativeQA (2017), which is no longer considered a challenging benchmark for modern RAG systems and may not be representative of common GraphRAG application scenarios. The authors should supplement their experiments with more complex, practical use cases where GraphRAG is specifically needed and standard RAG would be insufficient. Otherwise, the practical significance of this work is questionable.

W3. The method's design appears to be more of an engineering effort and lacks principled innovation. The main contributions seem to be demonstrating the value of operations like deduplication and clustering. However, these are standard pre-processing techniques in graph manipulation and can hardly be considered novel contributions of this work.

W4. The practical utility of the proposed method is ambiguous. Once the importance scores for entities and relations are obtained, what is their explicit use case? Can they be used for debugging the knowledge graph, refining the retrieval process, or improving model factuality? The authors need to clearly articulate the downstream applications and benefits of their method.

**Questions:**

In addition to the points in the "Weaknesses" section, I have the following questions:

Q1. The knowledge graph (KG) is a relatively traditional method of information representation and has several inherent limitations. For instance, it is challenging to represent complex content (such as the equations in your paper) using a KG, whereas text can be seen as a more general-purpose information carrier. In fact, most existing KGs are constructed by extracting information from text. This raises a fundamental question: why do we need to revert from text back to the structure of a knowledge graph?
Historically, due to the limitations in text processing and representation capabilities, KGs were utilized to simplify textual information and remove redundancy, thereby facilitating better representation and retrieval. However, modern Large Language Models (LLMs) can now effortlessly handle the complexities of text representation. Therefore, I believe the remaining advantages of knowledge graphs in the current era of LLMs warrant a more in-depth analysis and discussion.

Q2. I observed in your experiments that the effectiveness of your proposed method is primarily benchmarked against RAG-Ex. However, your method utilizes a graph structure, while RAG-Ex operates directly on text. Does this comparison introduce a potential unfairness in terms of the data modality? After all, the conversion of text into a graph is a necessary prerequisite for any knowledge graph-based approach. Consequently, it is difficult to consider the mere utilization of a graph structure as a novel contribution of your method. Could you elaborate on this?

---

> ### Author Response · Authors · 2025-12-02
> **#1 Knowledge Graphs and Text based content with LLMs**
>
> We thank the reviewer for raising this fundamental and insightful question regarding the role of Knowledge Graphs in the era of LLMs. We agree that while LLMs have revolutionised text processing, the structural advantages of KGs remain critical, particularly for the specific goals of our proposed method, XGRAG.
>
> There are distinct advantages KGs offer over raw text like discrete units for causal analysis, this is the core motivation for XGRAG, to achieve explainability via perturbation, we require clean, semantically meaningful units (nodes/edges). Perturbing raw text is often noisy and ambiguous, whereas removing a specific node allows for a precise analysis of its contribution to the final answer. Also, explicit structure for faithfulness, the symbolic structure of a KG helps ground the LLM, serving as a verification mechanism to reduce hallucination compared to purely implicit reasoning over unstructured text. Finally, efficiency, KGs act as a pre-computed index, allowing for the retrieval of small, highly relevant subgraphs, which is more computationally efficient than processing entire documents in long-context windows.
>
> We believe this discussion clarifies that while LLMs excel at understanding text, KGs provide the necessary structured base for verifiable reasoning and precise explainability.

---

> ### Author Response · Authors · 2025-12-02
> **#2 RAG-EX and XGRAX Fairness and Novelty**
>
> We thank the reviewer  for this insightful question, which touches upon the core motivation and contribution of our work. We'd like to elaborate on why this comparison is not only fair but essential for validating our approach, and clarify where the novelty of our method lies.
>
> **1. Fairness.** Experiments are designed to compare two different explanation strategies for Retrieval-Augmented Generation (RAG) systems, a text-native versus our graph-native one. We evaluate the effectiveness of an explanation system designed for GraphRAG based retrieval (XGRAG) against standard text-RAG (RAG-Ex). Since the primary motivation for GraphRAG itself is to leverage structured knowledge, it is logical and necessary that its corresponding explanation method should also operate on that structure. The experiment demonstrates that the structural benefits of a KG for retrieval also extend to explainability.
>
> **2. Novelty.** The core novelty of XGRAG lies in the design and validation of graph-native perturbation strategies and the demonstration that these strategies provide a superior analysis in XAI.
> - Text-based perturbation, is inherently ambiguous. Removing a sentence or a chunk of text is a coarse and noisy operation. A single sentence may contain multiple distinct facts, and its removal can disrupt narrative flow in unintended ways, making it difficult to isolate the precise cause of a change in the model's output.
> - Graph-based perturbation is surgically precise. Our method operates on semantically discrete units. Removing or altering a single entity or an edge is a clean, targeted intervention. This allows us to ask highly specific causal questions. This level of precision is simply not possible with text-based methods.
>
> Therefore, the reversion from text to a KG is the very step that enables our contribution. We convert text into a structured using state-of-the-art technology (LightRAG), symbolic map of knowledge precisely because that map provides the clean, discrete components necessary for a rigorous, perturbation-based causal analysis. The novelty is the methodology that leverages this structure to produce more faithful and fine-grained explanations, a capability that we demonstrate is superior to the current state-of-the-art in text-based RAG explanation. XGRAG can be applied to several fields as high-stakes domains, where the cost of an incorrect or unfaithful answer is high, our framework can be used to audit the model's reasoning process, ensuring that its conclusions are based on valid evidence.
> For developers, XGRAG serves as a powerful debugging tool, allowing them to pinpoint the specific knowledge graph components that may be leading to incorrect answers or hallucinations.
> Finally, for end-users, it fosters trust and transparency by providing a clear and understandable justification for the model's output, turning the "black box" of the LLM's reasoning into a transparent and auditable process.

---

> ### Author Response · Authors · 2025-12-02
> **#W2 Dataset Expansion**
>
> We thank the reviewer for raising the concern of insufficient datasets.
>
> In response, we have expanded our evaluation to include two additional datasets: FairytaleQA (Xu et al., 2022), a reading comprehension benchmark based on fairy tale stories and TriviaQA (Joshi et al., 2017), a large-scale QA dataset featuring questions from trivia domains and designed to test factual and contextual understanding. We conducted our experiments on these two newly added datasets using the same experimental settings as for NarrativeQA. For both FairytaleQA and TriviaQA, we also categorized the data into different story and question types and aggregated the results accordingly. This consistent evaluation protocol ensures comparability and further supports the robustness of our findings across diverse domains.
>
> The following table summarizes the experimental results on these two newly added datasets. For RAG-Ex, we conducted experiments at both word-level and sentence-level, while for XGRAG, we evaluated performance at node-level and edge-level. The reported results represent averages across all stories and question types.
>
> |   **Method**    |   **Granularity** | **F1**   |   **MRR**   |    **P@10%**   |   **P@30%**   |   **P@50%**   |
> |----------------------|------------------------|----------|--------|------------|------------|------------|
> | RAG-Ex  | word-level | 0.45 | 0.09 | 0.03 | 0.09 | 0.18
> | RAG-Ex  | sentence-level | 0.34 | **0.84** | 0.50 | **0.44** | 0.53
> | XGRAG (ours) | node-level | **0.48** | 0.64 | **0.52** | 0.43 | **0.56**
> | XGRAG (ours)  | edge-level | 0.44 | 0.48 | 0.13 | 0.33 | 0.42
>
> *Table 1: Experimental results on FairytaleQA and TriviaQA.*
>
> The results indicate that XGRAG consistently outperforms RAG-Ex across most metrics, particularly at node-level granularity, where it achieves the highest F1 and strong precision scores. While RAG-Ex performs competitively at sentence-level for MRR, XGRAG demonstrates greater overall robustness, confirming the effectiveness of graph-based reasoning for diverse QA tasks.
> The results are further aggregated with the previous results and the NarrativeQA dataset and are shown as follows:
>
> |   **Method**    |   **Granularity** | **F1**   |   **MRR**   |    **P@10%**   |   **P@30%**   |   **P@50%**   |
> |----------------------|------------------------|----------|--------|------------|------------|------------|
> | RAG-Ex  | word-level | 0.54 | 0.23 | 0.08 | 0.11 | 0.19
> | RAG-Ex  | sentence-level | 0.34 | 0.61 | 0.35 | 0.42 | 0.54
> | XGRAG (ours) | node-level | **0.62** | **0.72** | **0.66** | **0.44** | **0.57**
> | XGRAG (ours)  | edge-level | 0.52 | 0.65 | 0.22 | 0.42 | 0.48
>
> *Table 2: Experimental results on NarrativeQA, FairytaleQA and TriviaQA.*

---

### Official Review · Reviewer_vcFv · 2025-10-30

**Soundness:** 2
**Presentation:** 3
**Contribution:** 2
**Rating:** 2
**Confidence:** 4

**Summary:**

The paper proposes XGRAG, an explainable framework designed to identify the most influential nodes and edges that contributes to the outputs of GraphRAG models. The XGRAG framework consists of four components: a GraphRAG model for indexing and retrieving relevant subgraphs, an entity deduplication module for consolidating semantically similar entities, a perturber that systematically perturbs subgraphs, and an explaner that quantifies the importance of individual graph components to the model’s final response.

**Strengths:**

1. The paper proposes XGRAG to generate fine-grained explanations that identify the most influential nodes and edges contributing to an LLM’s response.
2. The paper conducts experiments on the NarrativeQA dataset and the results indicate the the proposed XGRAG outperforms a baseline RAG-Ex.
3. The paper is well-written and easy to follow.

**Weaknesses:**

1. The paper is incremental since the main idea of XGRAG, i.e., perturbing retrieved content to identify the most important components, has already been explored in previous works like RAG-Ex.
2. The paper only compares against RAG-Ex, which is designed for text-based RAG models, and lacks comprehensive comparisons with more relevant baselines, such as KGRAG-Ex.
3. The paper lacks sufficient justification for the proposed approach. For example, a simple baseline could involve using GraphRAG to generate predictions and then computing the similarity between graph components and the predicted answer to identify important elements. Additionally, the paper notes that XGRAG's importance scores align with structural properties such as Degree and PageRank. This raises the question of why these existing graph metrics are not used directly for identifying influential components.
4. The proposed method raises efficiency concerns, as it requires perturbing each graph component and generating outputs with the LLM. However, the paper does not provide statistics on the size of the retrieved subgraphs, making it difficult to assess the practical efficiency and scalability of the approach.
5. The paper only conducts experiments on the NarrativeQA dataset, raising concerns about whether the proposed approach can generalise to other datasets or domains.

**Questions:**

1. How does XGRAG differ from prior work like RAG-Ex and KGRAG-Ex, beyond being applied to GraphRAG?
2. Why are graph-based baselines such as KGRAG-Ex not included in the experimental comparison?
3. Can the authors provide an efficiency analysis to quantify the computational cost of XGRAG, particularly given the need to perturb and repeatedly invoke the LLM for each graph component?

---

> ### Author Response · Authors · 2025-12-02
> **#1 XGRAG Contributions**
>
> We thank the reviewer for prompting a detailed comparison between XGRAG and KGRAG-Ex, which allows us to clarify the key distinctions and advantages of our approach:
>
> **XGRAG vs. RAG-Ex**
> RAG-Ex applies perturbations to plain text paragraphs, identifying influential text segments for answer generation. In contrast, XGRAG operates on graph components (nodes & edges) within a knowledge graph. This shift from text to graph structure enables XGRAG to capture how graph topology and connectivity influence retrieval and reasoning, providing insights beyond textual importance. Importantly, our additional experiments on graph structure further demonstrate that answer generation is implicitly affected by graph connectivity, reinforcing the need for a graph-centric evaluation approach beyond text-only perturbations.
>
> **XGRAG vs. KGRAG-Ex**
> XGRAG advances beyond KGRAG-Ex in twofold. First, while KGRAG-Ex also introduced graph-level perturbations for explainability, it lacks a dedicated evaluation pipeline and does not report standardised metrics for assessing explanation quality, making fair comparison and reproducibility difficult.  This limitation also undermines the plausibility of their explainability results, as there is no objective basis for evaluating or trusting the explanations produced. In contrast, XGRAG systematically quantifies explanation accuracy using established metrics such as F1 and MRR, and aligns its evaluation with ground truth derived from semantic similarity to the model’s answers.  We have conducted extensive experiments, including direct comparisons against strong baselines like RAG-Ex, to rigorously demonstrate the superiority of our approach. Therefore, we believe our results are not only more robust but also more plausible and trustworthy than KGRAG-Ex. Furthermore, unlike KGRAG-Ex, XGRAG fully leverages the superior retrieval and graph construction strategies of LightRAG, further enhancing both the scalability and effectiveness of our explainability framework.
>
> XGRAG methodology leverages this structure to produce more faithful and fine-grained explanations, a capability that we demonstrate is superior to the current state-of-the-art in RAG-EX explanation. XGRAG can be applied to several fields as high-stakes domains, where the cost of an incorrect or unfaithful answer is high, our framework can be used to audit the model's reasoning process, ensuring that its conclusions are based on valid evidence.
> For developers, XGRAG serves as a powerful debugging tool, allowing them to pinpoint the specific knowledge graph components that may be leading to incorrect answers or hallucinations.
> Finally, for end-users, it fosters trust and transparency by providing a clear and understandable justification for the model's output, turning the "black box" of the LLM's reasoning into a transparent and auditable process.

---

> ### Author Response · Authors · 2025-12-02
> **#2 XGRAG Baseline Comparison**
>
> We thank the reviewer for raising this important point. We acknowledge that KGRAG-Ex (Balanos et al., 2025) presents a related graph-based perturbation pipeline designed to identify influential context pieces for answer generation, making it a natural candidate for comparison.
>
> While we considered KGRAG-Ex as a potential baseline, several practical constraints prevented its inclusion in our experiments:
>
> **1. Lack of Standardised Evaluation Metrics.** KGRAG-Ex does not provide standardised evaluation metrics in their implementation, nor does the paper offer a clear description of how outputs were assessed. This absence makes a fair and meaningful comparison infeasible without substantial reimplementation effort.
>
> **2. Absence of Reproducible Baseline Results.** The original work does not report baseline results that could be directly reused or adapted to our experimental setting. This further complicates integration into our evaluation framework.
>
> Given these limitations, we opted to focus on baselines for which reproducible implementations and well-defined evaluation were available, ensuring a rigorous and fair comparison. We agree that a direct comparison with KGRAG-Ex could strengthen future work and welcome collaboration or additional resources from the community to facilitate such evaluation.

---

> ### Author Response · Authors · 2025-12-02
> **#3 XGRAG Scalability and Performance Analysis**
>
> We thank the reviewers for raising this concern. We performed an additional analysis to determine scalability and performance for XGRAG.
>
> (1) We utilize the performance analysis done by the LightRAG (Guo et al., 2024), where they illustrate the
> comparative cost of a single query retrieval, based on the analysis by Guo et al. (2024).
>  $C_{max}$  is the maximum tokens per API call.
>
> | **Backbone** | **Token Load**      | **API Calls**                |
> |--------------|----------------------|------------------------------|
> | GraphRAG     | ≈ 610,000           | ≈ 610,000 / $C_{max}$    |
> | LightRAG     | < 100               | 1                            |
>
>
> (2) Formalisation of the XGRAG Cost.
>  $$ C_{XGRAG} = C_{baseline} + \sum_{p \in P} C_{p} \approx  N_p \times C_{invoke} $$
> where $N_p = N_{entities} \vee N_{edges}$, which is strategy-dependent, and $C_{invoke}$ is the cost of a single retrieval backbone invocation.
>
> (3) Compression between XGRAG_GraphRAG and XGRAG_LightRAG Cost.
>
> By substituting $C_{invoke}$ with the costs reported by (Guo et al., 2024), the advantage becomes clear, where GraphRAG backbone would be computationally infeasible, and LightRAG's  cost remains manageable.
>
> (3.1) $C_{XGRAG-GraphRAG} \propto N_p \times (\text{100s of API calls} + 10^5\text{-}10^6 \text{ tokens}) $
>
> (3.2) $ C_{XGRAG-LightRAG} \propto N_p \times (\text{1 API call} + 10^2 \text{ tokens}) $
>
> Finally, this analysis demonstrates that while our perturbation strategies multiplies the cost of a single query (2), the efficiency of LightRAG (1) is the key enabling factor that makes XGRAG a practical and scalable solution for explaining graph-based RAG (3.2).

---

> ### Author Response · Authors · 2025-12-02
> **#W5 Dataset Expansion**
>
> We thank the reviewer for raising the concern of generalizability.
>
> In response, we have expanded our evaluation to include two additional datasets: FairytaleQA (Xu et al., 2022), a reading comprehension benchmark based on fairy tale stories and TriviaQA (Joshi et al., 2017), a large-scale QA dataset featuring questions from trivia domains and designed to test factual and contextual understanding. We conducted our experiments on these two newly added datasets using the same experimental settings as for NarrativeQA. For both FairytaleQA and TriviaQA, we also categorized the data into different story and question types and aggregated the results accordingly. This consistent evaluation protocol ensures comparability and further supports the robustness of our findings across diverse domains.
>
> The following table summarizes the experimental results on these two newly added datasets. For RAG-Ex, we conducted experiments at both word-level and sentence-level, while for XGRAG, we evaluated performance at node-level and edge-level. The reported results represent averages across all stories and question types.
>
> |   **Method**    |   **Granularity** | **F1**   |   **MRR**   |    **P@10%**   |   **P@30%**   |   **P@50%**   |
> |----------------------|------------------------|----------|--------|------------|------------|------------|
> | RAG-Ex  | word-level | 0.45 | 0.09 | 0.03 | 0.09 | 0.18
> | RAG-Ex  | sentence-level | 0.34 | **0.84** | 0.50 | **0.44** | 0.53
> | XGRAG (ours) | node-level | **0.48** | 0.64 | **0.52** | 0.43 | **0.56**
> | XGRAG (ours)  | edge-level | 0.44 | 0.48 | 0.13 | 0.33 | 0.42
>
> *Table 1: Experimental results on FairytaleQA and TriviaQA.*
>
> The results indicate that XGRAG consistently outperforms RAG-Ex across most metrics, particularly at node-level granularity, where it achieves the highest F1 and strong precision scores. While RAG-Ex performs competitively at sentence-level for MRR, XGRAG demonstrates greater overall robustness, confirming the effectiveness of graph-based reasoning for diverse QA tasks.
> The results are further aggregated with the previous results and the NarrativeQA dataset and are shown as follows:
>
> |   **Method**    |   **Granularity** | **F1**   |   **MRR**   |    **P@10%**   |   **P@30%**   |   **P@50%**   |
> |----------------------|------------------------|----------|--------|------------|------------|------------|
> | RAG-Ex  | word-level | 0.54 | 0.23 | 0.08 | 0.11 | 0.19
> | RAG-Ex  | sentence-level | 0.34 | 0.61 | 0.35 | 0.42 | 0.54
> | XGRAG (ours) | node-level | **0.62** | **0.72** | **0.66** | **0.44** | **0.57**
> | XGRAG (ours)  | edge-level | 0.52 | 0.65 | 0.22 | 0.42 | 0.48
>
> *Table 2: Experimental results on NarrativeQA, FairytaleQA and TriviaQA.*

---

### Official Review · Reviewer_Uj7w · 2025-10-30

**Soundness:** 1
**Presentation:** 3
**Contribution:** 3
**Rating:** 2
**Confidence:** 4

**Summary:**

This paper aims to explain the outputs of Graph-based Retrieval-Augmented Generation (GraphRAG) systems. The authors propose a perturbation-based explanation approach, extended from existing work RAG-EX. The proposed method XGRAG uses LightRAG as the backbone and introduces graph-based perturbations such as node and edge removal. The experiments include only one baseline and one dataset.

**Strengths:**

1. The paper is well-written and clearly articulates its core idea.
2. The proposed approach reasonably integrates LightRAG with RAG-EX, introducing two key modifications: improved entity deduplication and a graph perturbation mechanism.

**Weaknesses:**

1. The experimental evaluation lacks comprehensiveness, with only one dataset and one baseline included.
2. An important evaluation relies on a hypothesis that is neither convincingly argued nor adequately justified in the paper.

**Questions:**

1. The evaluation includes only one baseline, RAG-Ex. The authors should justify why KGRAG-Ex (Balanos et al., 2025) was not included as a baseline comparison.
2. The use of a single dataset for evaluation undermines the generalizability of the experimental results.
3. Figures 3 and 4 employ an inappropriate visualization method: line plots are used despite the x-axis representing discrete variables, when bar plots would be the correct choice for non-continuous data.
4. In Table 3, the purpose of evaluating across different LLMs is to demonstrate that the conclusion, XGRAG outperforms RAG-Ex, remains consistent regardless of the LLM used. However, to validate this claim, the results for RAG-Ex should also be reported alongside XGRAG for each LLM, rather than showing only XGRAG performance.
5. The hypothesis stated in Line 339 is questionable. In KG retrieval, triples relevant to a query do not necessarily correspond to those with high structural importance. Furthermore, standard PageRank assumes homogeneous edge semantics and may not perform effectively on multi-relational KGs [A]. Given that this hypothesis is central to validating the quality of graph explanations, the authors should provide more rigorous justification and clarification.
6. In Line 450, the similarity threshold is not defined, nor is the selection methodology explained. Please provide this information.
7. The paper lacks error analysis. Specifically, the authors should investigate which questions can be successfully explained by RAG-Ex, XGRAG, both methods, or neither, and analyze the underlying reasons for these outcomes. Furthermore, the necessity of KGs for explanation should be empirically demonstrated through comparative analysis.

[A] Li, X., Ng, M. K., & Ye, Y. (2012, April). HAR: hub, authority and relevance scores in multi-relational data for query search. In Proceedings of the 2012 SIAM International Conference on Data Mining (pp. 141-152). Society for Industrial and Applied Mathematics.

---

> ### Author Response · Authors · 2025-12-02
> **#1 XGRAG Baseline Comparison**
>
> We thank the reviewer for raising this important point. We acknowledge that KGRAG-Ex (Balanos et al., 2025) presents a related graph-based perturbation pipeline designed to identify influential context pieces for answer generation, making it a natural candidate for comparison.
>
> While we considered KGRAG-Ex as a potential baseline, several practical constraints prevented its inclusion in our experiments:
>
> **1. Lack of Standardised Evaluation Metrics.** KGRAG-Ex does not provide standardised evaluation metrics in their implementation, nor does the paper offer a clear description of how outputs were assessed. This absence makes a fair and meaningful comparison infeasible without substantial reimplementation effort.
>
> **2. Absence of Reproducible Baseline Results.** The original work does not report baseline results that could be directly reused or adapted to our experimental setting. This further complicates integration into our evaluation framework.
>
> Given these limitations, we opted to focus on baselines for which reproducible implementations and well-defined evaluation were available, ensuring a rigorous and fair comparison. We agree that a direct comparison with KGRAG-Ex could strengthen future work and welcome collaboration or additional resources from the community to facilitate such evaluation.

---

> ### Author Response · Authors · 2025-12-02
> **#3 Figure Visualisation Method Change**
>
> We appreciate the feedback from the reviewer regarding the visualisation methods in Figures 3 and 4.
> You are correct that line plots are inappropriate for discrete variables on the x-axis, as they imply continuity between categories, which does not reflect the nature of the data. Bar plots are indeed the standard choice for representing non-continuous, categorical data, as they clearly distinguish between distinct categories without suggesting interpolation between them.
>
> In the revised version of the paper, we have replaced the line plots in Figures 3 and 4 with bar plots to ensure accurate and appropriate representation of the data. This adjustment enhances the clarity of our findings and aligns with best practices in data visualisation.

---

> ### Author Response · Authors · 2025-12-02
> **#5 KG Structural Relevance**
>
> We thank the reviewer for this thoughtful critique regarding our hypothesis in Line 339.
>
> **1. Clarification on Scope.** We would like to clarify that our structural importance analysis is not applied during the KG retrieval phase. All graph components subject to perturbation are already part of the subgraph retrieved for the given query. Our perturbation-based importance scoring operates exclusively within this retrieved context. The purpose of correlating explainer importance scores with structural centrality metrics (e.g., PageRank, Degree Centrality) is to assess intrinsic graph relevance, beyond purely textual relevance, during answer generation. Specifically, we investigate whether structurally central components within the retrieved subgraph exert greater influence on the model's output, given the context provided to the LLM.
>
> **2. On Structural Metrics in Multi-Relational KGs.** We acknowledge that standard PageRank assumes homogeneous edge semantics and may not fully capture the nuances of multi-relational knowledge graphs. In our work, PageRank and Degree Centrality serve as baseline structural metrics, providing an intrinsic perspective on the alignment between our explainer's importance scores and the underlying graph topology. These metrics might not express the full relevance on multi-relational setting, but they function as a verification that our explanation method remains sensitive to structural properties within the retrieved context.
>
> **3. Primary vs. Complementary Evaluation.** To further clarify, our primary evaluation of explanation quality is grounded in semantic relevance to the generated answer. The structural alignment analysis serves as a complementary, intrinsic validation rather than the central evaluation criterion. We agree that future work should explore relation-aware centrality measures specifically designed for multi-relational graphs, which may yield a more nuanced assessment of structural importance in such settings.

---

> ### Author Response · Authors · 2025-12-02
> **#6 Entity Deduplication Sensitivity Analysis**
>
> We appreciate the reviewer's point, the Entity Deduplication Threshold can be found in Appendix D.3 Hyperparameters. Additionally, we have added the section D.4 Entity Deduplication Sensitivity Analysis, which demonstrates that $\theta_{sim}=0.7$ provides the best balance between ranking quality and overall performance.
>
> (1) Performance Stability Across Thresholds
> Performance remains relatively stable for low thresholds but varies more at mid and high ranges. While $\theta_{sim}=0.7$ has a slightly lower F1 score (0.794 compared to 0.828 at very low thresholds), it consistently outperforms in RR and ranking overlap metrics (top10%, top30%, top50%), making it the best balance.
>
> |   **$\theta_{sim}$**    |   **F1**   |   **MRR**   |    **P@10%**   |   **P@30%**   |   **P@50%**   |
> |-----------------------|----------|--------|------------|------------|------------|
> | 0.00  | 0.828 | 0.9 | 0.8 | 0.8 | 0.7 |
> | 0.05  | 0.828 | 0.9 | 0.8 | 0.8 | 0.7 |
> | 0.10  | 0.828 | 0.9 | 0.8 | 0.8 | 0.7 |
> | 0.15  | 0.828 | 0.9 | 0.8 | 0.8 | 0.7 |
> | 0.20  | 0.828 | 0.9 | 0.8 | 0.8 | 0.7 |
> | 0.25  | 0.828 | 0.9 | 0.8 | 0.8 | 0.7 |
> | 0.30  | 0.828 | 0.9 | 0.8 | 0.8 | 0.594 |
> | 0.35  | 0.828 | 0.9 | 0.8 | 0.8 | 0.594 |
> | 0.40  | 0.828 | 1   | 0.8 | 0.8 | 0.61 |
> | 0.45  | 0.728 | 1   | 0.8 | 0.734 | 0.666 |
> | 0.50  | 0.728 | 1   | 0.8 | 0.734 | 0.666 |
> | 0.55  | 0.694 | 1   | 0.8 | 0.6 | 0.7 |
> | 0.60  | 0.694 | 1   | 0.8 | 0.6 | 0.7 |
> | 0.65  | 0.794 | 1   | 1   | 0.802 | 0.702 |
> | **0.70**  | **0.794** | **1** | **1** | **0.802** | **0.702** |
> | 0.75  | 0.794 | 1   | 1   | 0.802 | 0.702 |
> | 0.80  | 0.794 | 1   | 1   | 0.802 | 0.702 |
> | 0.85  | 0.794 | 1   | 1   | 0.802 | 0.702 |
> | 0.90  | 0.794 | 1   | 1   | 0.802 | 0.702 |
> | 0.95  | 0.794 | 1   | 0.8 | 0.6 | 0.57 |
> | 1.00  | 0.794 | 1   | 0.8 | 0.6 | 0.57 |
>
> Table **Sensitivity analysis of similarity threshold $\theta_{sim}$ on entity deduplication performance**
> *(Results based on experiments using `llava-7b` with the "Goldilocks and the Three Bears" story as input.)*
>
> (2) Justification for $\theta_{sim}=0.7$
> This threshold avoids the mid-range dip in F1 and aligns with the region where ranking quality is maximised, ensuring strong retrieval consistency while maintaining acceptable precision and recall.

---

> ### Author Response · Authors · 2025-12-02
> **#7 Question Type Analysis for RAG-EX and XGRAG**
>
> Thank you for highlighting the need for error analysis. We agree that understanding which questions are successfully explained by RAG-Ex, XGRAG, both, or neither is essential. In our current work, we have partially addressed this by dividing the questions according to narrative types, which allowed us to observe how the proposed solution performs across different story structures. This categorisation provides initial insights into performance variations.  The proposed framework achieves its highest effectiveness on simple and abstract narratives, while its performance degrades when applied to stories with more complex plot.
>
> We acknowledge that a deeper investigation, such as identifying the underlying reasons for success or failure for each method and empirically demonstrating the necessity of KGs is valuable.

---

> ### Author Response · Authors · 2025-12-02
> **#2 Dataset Expansion**
>
> We thank the reviewer for highlighting the importance of generalizability.
>
> In response, we have expanded our evaluation to include two additional datasets: FairytaleQA (Xu et al., 2022), a reading comprehension benchmark based on fairy tale stories and TriviaQA (Joshi et al., 2017), a large-scale QA dataset featuring questions from trivia domains and designed to test factual and contextual understanding. We conducted our experiments on these two newly added datasets using the same experimental settings as for NarrativeQA. For both FairytaleQA and TriviaQA, we also categorized the data into different story and question types and aggregated the results accordingly. This consistent evaluation protocol ensures comparability and further supports the robustness of our findings across diverse domains.
>
> The following table summarizes the experimental results on these two newly added datasets. For RAG-Ex, we conducted experiments at both word-level and sentence-level, while for XGRAG, we evaluated performance at node-level and edge-level. The reported results represent averages across all stories and question types.
>
> |   **Method**    |   **Granularity** | **F1**   |   **MRR**   |    **P@10%**   |   **P@30%**   |   **P@50%**   |
> |----------------------|------------------------|----------|--------|------------|------------|------------|
> | RAG-Ex  | word-level | 0.45 | 0.09 | 0.03 | 0.09 | 0.18
> | RAG-Ex  | sentence-level | 0.34 | **0.84** | 0.50 | **0.44** | 0.53
> | XGRAG (ours) | node-level | **0.48** | 0.64 | **0.52** | 0.43 | **0.56**
> | XGRAG (ours)  | edge-level | 0.44 | 0.48 | 0.13 | 0.33 | 0.42
>
> *Table 1: Experimental results on FairytaleQA and TriviaQA.*
>
> The results indicate that XGRAG consistently outperforms RAG-Ex across most metrics, particularly at node-level granularity, where it achieves the highest F1 and strong precision scores. While RAG-Ex performs competitively at sentence-level for MRR, XGRAG demonstrates greater overall robustness, confirming the effectiveness of graph-based reasoning for diverse QA tasks.
> The results are further aggregated with the previous results and the NarrativeQA dataset and are shown as follows:
>
> |   **Method**    |   **Granularity** | **F1**   |   **MRR**   |    **P@10%**   |   **P@30%**   |   **P@50%**   |
> |----------------------|------------------------|----------|--------|------------|------------|------------|
> | RAG-Ex  | word-level | 0.54 | 0.23 | 0.08 | 0.11 | 0.19
> | RAG-Ex  | sentence-level | 0.34 | 0.61 | 0.35 | 0.42 | 0.54
> | XGRAG (ours) | node-level | **0.62** | **0.72** | **0.66** | **0.44** | **0.57**
> | XGRAG (ours)  | edge-level | 0.52 | 0.65 | 0.22 | 0.42 | 0.48
>
> *Table 2: Experimental results on NarrativeQA, FairytaleQA and TriviaQA.*

---

### Official Review · Reviewer_Hi18 · 2025-10-31

**Soundness:** 3
**Presentation:** 2
**Contribution:** 2
**Rating:** 4
**Confidence:** 3

**Summary:**

This paper proposes XGRAG, an explainability framework for Graph-based Retrieval-Augmented Generation (GraphRAG) systems. Unlike previous explainability methods that operate on unstructured text, XGRAG performs graph-native perturbations — removing nodes, edges, or injecting synonyms — to quantify the causal influence of each graph component on the final LLM answer. The framework integrates (1) entity deduplication to merge semantically equivalent nodes, (2) perturb-generate-compare evaluation to compute importance scores, and (3) alignment with graph structural measures to assess validity. Experiments on NarrativeQA show that XGRAG improves explanation accuracy, robustness across story types and question complexities, and generalization across multiple open-source LLMs (LLaMA 3.1-8B, Mistral-7B, LLaVA-7B, etc.). Ablation studies confirm that entity deduplication and node-level perturbations are key to performance gains.

**Strengths:**

1. Clear motivation. The paper identifies a genuine gap: existing XAI methods for RAG cannot interpret reasoning grounded in structured graph data. XGRAG directly addresses this with a graph-native perturbation approach.

2. Novel methods to perturb the graph. The “Perturb-Generate-Compare” paradigm is adapted elegantly to graphs, combining semantic and structural importance in a unified explanation measure.

3. Comprehensive experiments. Authors include strong empirical validation spans multiple LLMs, question types, and story structures. Ablation studies show the necessity of entity deduplication and tests three perturbation strategies.

**Weaknesses:**

1. Limited experimental domain. All experiments are conducted on NarrativeQA, evaluation on other domains (scientific, biomedical or factual QA/KGs) will better demonstrate the ability to generalize.

2. Potential biased evaluation. When building the ground truth, the authors take the assumption that "graph components semantically similar to the final answer are the most relevant pieces of evidence." This assumption can be ungrounded especially when there is no exact information that can directly solve the query, relevant (semantically similar) information in this case could cause hallucination [1].

3. Scalability issues. The framework requires multiple GraphRAG invocations per perturbation. While LightRAG mitigates some cost, scalability to very large KGs or multi-hop queries remains uncertain.

[1] GIVE: Structured Reasoning of Large Language Models with Knowledge Graph Inspired Veracity Extrapolation

**Questions:**

1. How does XGRAG scale to industrial-scale KGs (millions of entities)? Would LightRAG still be computationally feasible for large perturbation batches?

2. The evaluation relies on similarity to the final answer. Have the authors considered other evaluation metrics such as task-specific annotation to confirm faithfulness?

3. How sensitive is performance to the similarity threshold (θ_sim) used in entity deduplication? Can the authors include an ablation study for that?

---

> ### Author Response · Authors · 2025-12-02
> **#1 XGRAG Scalability and Performance Analysis**
>
> We thank the reviewers for raising this concern. We performed an additional analysis to determine scalability and performance for XGRAG.
>
> (1) We utilize the performance analysis done by the LightRAG (Guo et al., 2024), where they illustrate the
> comparative cost of a single query retrieval, based on the analysis by Guo et al. (2024).
>  $C_{max}$  is the maximum tokens per API call.
>
> | **Backbone** | **Token Load**      | **API Calls**                |
> |--------------|----------------------|------------------------------|
> | GraphRAG     | ≈ 610,000           | ≈ 610,000 / $C_{max}$    |
> | LightRAG     | < 100               | 1                            |
>
>
> (2) Formalisation of the XGRAG Cost.
>  $$ C_{XGRAG} = C_{baseline} + \sum_{p \in P} C_{p} \approx  N_p \times C_{invoke} $$
> where $N_p = N_{entities} \vee N_{edges}$, which is strategy-dependent, and $C_{invoke}$ is the cost of a single retrieval backbone invocation.
>
> (3) Compression between XGRAG_GraphRAG and XGRAG_LightRAG Cost.
>
> By substituting $C_{invoke}$ with the costs reported by (Guo et al., 2024), the advantage becomes clear, where GraphRAG backbone would be computationally infeasible, and LightRAG's  cost remains manageable.
>
> (3.1) $C_{XGRAG-GraphRAG} \propto N_p \times (\text{100s of API calls} + 10^5\text{-}10^6 \text{ tokens}) $
>
> (3.2) $ C_{XGRAG-LightRAG} \propto N_p \times (\text{1 API call} + 10^2 \text{ tokens}) $
>
> Finally, this analysis demonstrates that while our perturbation strategies multiplies the cost of a single query (2), the efficiency of LightRAG (1) is the key enabling factor that makes XGRAG a practical and scalable solution for explaining graph-based RAG (3.2).

---

> > ### Author Response · Authors · 2025-12-02
> > **#2 Faithfulness of Evaluation Metric**
> >
> > We thank the reviewer for this insightful suggestion regarding the robustness of our evaluation metrics. We did consider task-specific human annotation; however, we opted for semantic similarity as a proxy primarily to address scalability and efficiency.
> >
> > While we acknowledge that human annotation is the gold standard for assessing faithfulness, it is resource-intensive and difficult to scale. Our automated approach significantly reduces the human effort and costs associated with manual scoring, allowing us to benchmark the system across a larger volume of data than would be feasible with human evaluators alone. To mitigate this risk, we restricted our evaluation to instances where the model produced the correct answer, thereby minimising the assessment of explanations for hallucinated content.
> >
> > We agree, however, that a more direct evaluation is necessary for future development. Accordingly, we have updated the Future Work section of the paper to explicitly acknowledge this limitation. We have mentioned the incorporation of semi-automated methods, such as using advanced LLMs as judges, and human-annotated relevance scores in future iterations to provide a more rigorous confirmation of faithfulness.

---

> ### Author Response · Authors · 2025-12-02
> **#3 Entity Deduplication Sensitivity Analysis**
>
> (1) Performance Stability Across Thresholds
> Performance remains relatively stable for low thresholds but varies more at mid and high ranges. While $\theta_{sim}=0.7$ has a slightly lower F1 score (0.794 compared to 0.828 at very low thresholds), it consistently outperforms in MRR and ranking overlap metrics (P@10%, P@30%, P@50%), making it the best balance.
>
> |   **$\theta_{sim}$**    |   **F1**   |   **MRR**   |    **P@10%**   |   **P@30%**   |   **P@50%**   |
> |-----------------------|----------|--------|------------|------------|------------|
> | 0.00  | 0.828 | 0.9 | 0.8 | 0.8 | 0.7 |
> | 0.05  | 0.828 | 0.9 | 0.8 | 0.8 | 0.7 |
> | 0.10  | 0.828 | 0.9 | 0.8 | 0.8 | 0.7 |
> | 0.15  | 0.828 | 0.9 | 0.8 | 0.8 | 0.7 |
> | 0.20  | 0.828 | 0.9 | 0.8 | 0.8 | 0.7 |
> | 0.25  | 0.828 | 0.9 | 0.8 | 0.8 | 0.7 |
> | 0.30  | 0.828 | 0.9 | 0.8 | 0.8 | 0.594 |
> | 0.35  | 0.828 | 0.9 | 0.8 | 0.8 | 0.594 |
> | 0.40  | 0.828 | 1   | 0.8 | 0.8 | 0.61 |
> | 0.45  | 0.728 | 1   | 0.8 | 0.734 | 0.666 |
> | 0.50  | 0.728 | 1   | 0.8 | 0.734 | 0.666 |
> | 0.55  | 0.694 | 1   | 0.8 | 0.6 | 0.7 |
> | 0.60  | 0.694 | 1   | 0.8 | 0.6 | 0.7 |
> | 0.65  | 0.794 | 1   | 1   | 0.802 | 0.702 |
> | **0.70**  | **0.794** | **1** | **1** | **0.802** | **0.702** |
> | 0.75  | 0.794 | 1   | 1   | 0.802 | 0.702 |
> | 0.80  | 0.794 | 1   | 1   | 0.802 | 0.702 |
> | 0.85  | 0.794 | 1   | 1   | 0.802 | 0.702 |
> | 0.90  | 0.794 | 1   | 1   | 0.802 | 0.702 |
> | 0.95  | 0.794 | 1   | 0.8 | 0.6 | 0.57 |
> | 1.00  | 0.794 | 1   | 0.8 | 0.6 | 0.57 |
>
> Table **Sensitivity analysis of similarity threshold $\theta_{sim}$ on entity deduplication performance**
> *(Results based on experiments using `llava-7b` with the "Goldilocks and the Three Bears" story as input.)*
>
> (2) Justification for $\theta_{sim}=0.7$
> This threshold avoids the mid-range dip in F1 and aligns with the region where ranking quality is maximised, ensuring strong retrieval consistency while maintaining acceptable precision and recall.
>
> This analysis has been added in the manuscript under the appendix D.4 Entity Deduplication Sensitivity Analysis.

---

> ### Author Response · Authors · 2025-12-02
> **#W1 Dataset Expansion**
>
> We thank the reviewer for raising the concern of generalizability.
>
> In response, we have expanded our evaluation to include two additional datasets: FairytaleQA (Xu et al., 2022), a reading comprehension benchmark based on fairy tale stories and TriviaQA (Joshi et al., 2017), a large-scale QA dataset featuring questions from trivia domains and designed to test factual and contextual understanding. We conducted our experiments on these two newly added datasets using the same experimental settings as for NarrativeQA. For both FairytaleQA and TriviaQA, we also categorized the data into different story and question types and aggregated the results accordingly. This consistent evaluation protocol ensures comparability and further supports the robustness of our findings across diverse domains.
>
> The following table summarizes the experimental results on these two newly added datasets. For RAG-Ex, we conducted experiments at both word-level and sentence-level, while for XGRAG, we evaluated performance at node-level and edge-level. The reported results represent averages across all stories and question types.
>
> |   **Method**    |   **Granularity** | **F1**   |   **MRR**   |    **P@10%**   |   **P@30%**   |   **P@50%**   |
> |----------------------|------------------------|----------|--------|------------|------------|------------|
> | RAG-Ex  | word-level | 0.45 | 0.09 | 0.03 | 0.09 | 0.18
> | RAG-Ex  | sentence-level | 0.34 | **0.84** | 0.50 | **0.44** | 0.53
> | XGRAG (ours) | node-level | **0.48** | 0.64 | **0.52** | 0.43 | **0.56**
> | XGRAG (ours)  | edge-level | 0.44 | 0.48 | 0.13 | 0.33 | 0.42
>
> *Table 1: Experimental results on FairytaleQA and TriviaQA.*
>
> The results indicate that XGRAG consistently outperforms RAG-Ex across most metrics, particularly at node-level granularity, where it achieves the highest F1 and strong precision scores. While RAG-Ex performs competitively at sentence-level for MRR, XGRAG demonstrates greater overall robustness, confirming the effectiveness of graph-based reasoning for diverse QA tasks.
> The results are further aggregated with the previous results and the NarrativeQA dataset and are shown as follows:
>
> |   **Method**    |   **Granularity** | **F1**   |   **MRR**   |    **P@10%**   |   **P@30%**   |   **P@50%**   |
> |----------------------|------------------------|----------|--------|------------|------------|------------|
> | RAG-Ex  | word-level | 0.54 | 0.23 | 0.08 | 0.11 | 0.19
> | RAG-Ex  | sentence-level | 0.34 | 0.61 | 0.35 | 0.42 | 0.54
> | XGRAG (ours) | node-level | **0.62** | **0.72** | **0.66** | **0.44** | **0.57**
> | XGRAG (ours)  | edge-level | 0.52 | 0.65 | 0.22 | 0.42 | 0.48
>
> *Table 2: Experimental results on NarrativeQA, FairytaleQA and TriviaQA.*

---

### Meta-Review · Area_Chair_qcZX · 2026-01-03

**Summary:**

The paper proposes XGRAG, a perturbation-based framework to explain GraphRAG outputs by identifying influential nodes and edges. While the problem of explaining GraphRAG is well-motivated and the reviewers appreciated the clarity of the writing and the engineering integration with LightRAG, the submission cannot be accepted in its current form. The decision to reject is based on the unanimous consensus among reviewers (Scores: 4, 2, 2, 2). The primary grounds for rejection are the lack of appropriate graph-native baselines (comparing only against text-based RAG-Ex), concerns regarding the validity of the evaluation metric (ground truth based on semantic similarity), and limited technical novelty (viewed as an incremental engineering adaptation of existing perturbation methods).

**Reviewer Concerns:**

Concerns Addressed by Rebuttal:
- Dataset Scope: The authors successfully addressed the concern regarding limited experimental breadth (originally only NarrativeQA) by adding results for FairyTaleQA and TriviaQA during the rebuttal. This was a major point for Reviewers Hi18, Uj7w, and vcFv.
- Scalability/Cost: The authors provided a breakdown of the computational cost and justified the efficiency gains of using LightRAG over standard GraphRAG , addressing questions from Reviewers Hi18 and vcFv.
- Visualization: The authors corrected the inappropriate use of line plots for discrete variables, switching to bar plots as requested by Reviewer Uj7w.

Not Adequately Addressed:
- Baseline Selection: A critical, unresolved issue is the lack of graph-specific baselines. Reviewers Uj7w and vcFv pointed out that comparing XGRAG only against the text-based RAG-Ex is insufficient and creates a modality mismatch.
- Evaluation Validity: Reviewer Hi18 raised a fundamental concern that the "ground truth" is derived from semantic similarity to the final answer. This creates a circular logic where hallucinations that are semantically similar to the answer might be falsely validated as "faithful" explanations. The authors acknowledged this limitation but did not provide a robust alternative (e.g., human evaluation) during the rebuttal.
- Nature of "Explainability": Reviewer tvRz strongly argued that the method provides "importance attribution" or ranking rather than true explainability regarding the LLM's inner reasoning. The rebuttal did not fundamentally alter the method to address this conceptual gap.
- Incremental Novelty: Multiple reviewers viewed the work as an engineering application of existing perturbation techniques (from RAG-Ex) to a new data structure (Graphs), rather than a significant algorithmic innovation.

**Reviewer Scores:**

* Reviewer Hi18 (Initial Score: 4): The reviewer's score would likely remain at 4. While the rebuttal successfully addressed concerns regarding the experimental domain and scalability, the critical and fundamental concern about the validity of the evaluation metric (biased ground truth) was not resolved to the extent required for acceptance.

* Reviewer Uj7w (Initial Score: 2): The reviewer's score would likely remain at 2. Despite several specific questions being addressed (e.g., dataset expansion, visualization, theta_sim sensitivity, structural alignment clarification), the core weakness regarding the lack of comprehensive baseline comparisons, especially against other graph-native XAI methods like KGRAG-Ex, remained an outstanding and critical issue.

* Reviewer tvRz (Initial Score: 2): The reviewer's score would likely remain at 2. The conceptual issues concerning the true nature of "explainability" provided by the method and the perception of limited technical novelty were not fundamentally resolved by the rebuttal. These represent central criticisms for this reviewer, making a significant score increase improbable.

---

### Decision · Program_Chairs · 2026-01-26

Reject